# *Klebsiella oxytoca* facilitates microbiome recovery via antibiotic degradation and restores colonization resistance in a diet-dependent manner

Éva d. H. Almási [1], Lea Eisenhard[1,6], Lisa Osbelt [1,6], Till Robin Lesker[1], Anna C. Vetter [2], Nele Knischewski[1], Agata Anna Bielecka[1], Achim Gronow[1], Uthayakumar Muthukumarasamy[1], Marie Wende[1], Caroline Tawk[1], Meina Neumann-Schaal [3], Mark Brönstrup [2] & Till Strowig [1,4,5] ✉

Competition among bacteria for carbohydrates is pivotal for colonization resistance (CR). However, the impact of Western-style diets on CR remains unclear. Here we show how the competition between *Klebsiella oxytoca* and *Klebsiella pneumoniae* is modulated by consuming one of three Western-style diets characterized by high-starch, high-sucrose, or high-fat/high-sucrose content. In vivo competition experiments in ampicillin-treated mice reveal that *K. oxytoca* promotes *K. pneumoniae* decolonization on all dietary backgrounds. However, mice on the high-fat/high-sucrose diet show reduced pathogen clearance. Microbiome analysis reveals that the combination of Western-style diets and ampicillin treatment synergize in microbiome impairment, particularly noticeable in the presence of high dietary fat content. The diet-independent degradation of ampicillin in the gut lumen by *K. oxytoca* beta-lactamases facilitates rapid commensal outgrowth, which is required for subsequent pathogen clearance. Our findings provide insights into how diet modulates functional microbiome recovery and *K. oxytoca*-mediated pathogen elimination from the gut.

Even though the host encounters pathogens constantly, they rarely establish stable colonization in the body. Notably, a diverse gut microbiome contributes to protecting the host from invading pathogens under homeostatic conditions; this phenomenon is referred to as colonization resistance (CR)[1,2]. Gut commensals exclude pathogens, for example, as a result of the secretion of secondary metabolites[3–5], competition for electron acceptors[6,7], trace metals[8], and, importantly, nutrients[9–11]. Moreover, it has been described that intricate cross-feeding mechanisms occur between gut microbes to maximize nutrient utilization[12]. However, once CR is disrupted due to medical interventions such as antibiotic treatment or chemotherapy[13,14], the lack of microbial catabolism of nutrients enables pathogens to expand and take over previously occupied niches. Competition between pathogens and gut commensals for diverse nutrients such as β-glucosides[9], galactitol[10,11], fructose[15], and gluconate[16] has been described. Yet, in all of these studies, mice were fed chow diets, i.e., plant-derived high-fiber

[1]Department of Microbial Immune Regulation, Helmholtz Centre for Infection Research (HZI), Braunschweig, Germany. [2]Department of Chemical Biology, Helmholtz Centre for Infection Research (HZI), Braunschweig, Germany. [3]Bacterial Metabolomics, Leibniz Institute DSMZ-German Collection of Microorganisms and Cell Cultures, Braunschweig, Germany. [4]Center for Individualized Infection Medicine, Hannover, Germany. [5]German Center for Infection Research (DZIF), partner site Hannover-Braunschweig, Braunschweig, Germany. [6]These authors contributed equally: Lea Eisenhard, Lisa Osbelt. ✉e-mail: till.strowig@helmholtz-hzi.de

feed developed for laboratory rodents. But most diets consumed in developed countries – also referred to as Westernized diets – are products of complicated industrial processes resulting in serious deprivation of complex polysaccharides, yet, are rich in starches, mono- and disaccharides as well as animal-derived fats[17]. Long-term consumption of such diets has been linked to obesity, metabolic disease[18], inflammatory bowel disease[19,20], and reduced CR[21,22]. Specifically, increased susceptibility to infections with adherent-invasive *E. coli*[23], and reduced CR toward *Citrobacter rodentium*[24] were identified. Yet, the effects of the host diet on colonization resistance against the clinically highly relevant pathogen *Klebsiella pneumoniae* have only started to be investigated[25].

*K. pneumoniae* is a Gram-negative that is known to be a causative agent of nosocomial infections such as pneumonia, abdominal infections, bloodstream infections, and meningitis[26]. Since they use the gut as a reservoir[27], disturbed CR often leads to systemic infection in the individual[28]. As a result, there has been a growing effort in recent years to deploy targeted, commensal probiotics to tackle multidrug-resistant human pathogens[29,30]. For instance, *Klebsiella oxytoca* has been recently shown to decolonize nosocomial isolates of *K. pneumoniae* from the gut due to its extended and largely overlapping carbon-source utilization potential[9]. However, co-colonization dynamics have only been investigated so far in standard chow diets and thus, how Westernized diets affect *K. pneumoniae* decolonization has remained uncharacterized.

In this study, we investigate the effects of three Westernized diets on the colonization of *K. pneumoniae* and *K. oxytoca* as well as their competitive dynamics in a mouse model. Our findings show that *K. oxytoca* reduces gut colonization of *K. pneumoniae* through direct

antagonism in the early phase of infection on all tested diets. However, subsequent complete pathogen gut decolonization is impaired in the presence of high-fat contents in the diet, albeit an effect of *K. oxytoca* also on this diet. Co-colonization of germ-free (GF) mice fed chow or a Western-style diet, which supports full clearance in specific-pathogen-free (SPF) mice, leads to the long-term co-existence of both strains highlighting the importance of co-colonizing gut commensals. Longitudinal microbiota analysis reveals compositional and functional aspects of gut microbiome recovery post-ampicillin treatment that are modulated by Western-style diets, including the identification of a microbiota signature associated with diet-independent decolonization. Finally, we also provide evidence that *K. oxytoca* further contributed to the re-establishment of CR by ampicillin degradation in the intestinal content, which enabled faster compositional and functional microbiota recovery.

## Results

### Western-style diets influence the kinetics of *K. oxytoca*-mediated clearance of *K. pneumoniae*

To assess the effects of different diets on *K. oxytoca* and *K. pneumoniae* colonization and competition, a previously established in vivo model was adapted[9] (Fig. 1a). SPF mice were fed either standard chow or one of three semi-synthetic diets: high-starch (HSt), high-sucrose (HS), or high-fat/high-sucrose (HF/HS), representing different Westernized diets[31]. Colonization resistance of mice was disrupted by ampicillin supplementation to the drinking water starting 48 hours after the beginning of the dietary intervention (2 days post-diet; dpd). On 5 dpd, one experimental group per diet was colonized with *K. oxytoca* MK01, while the other group was treated as the control group. High levels of

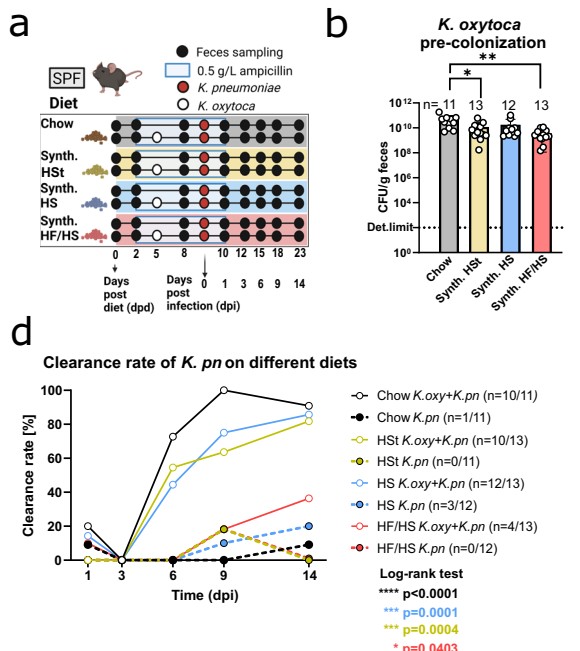

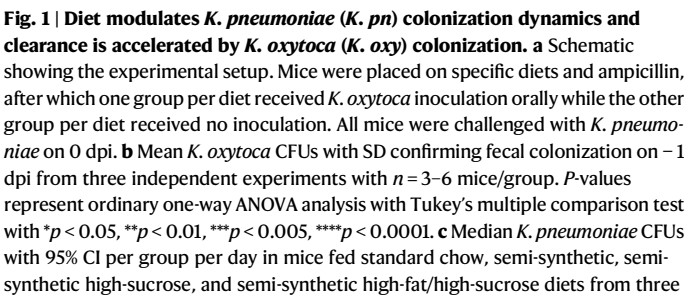

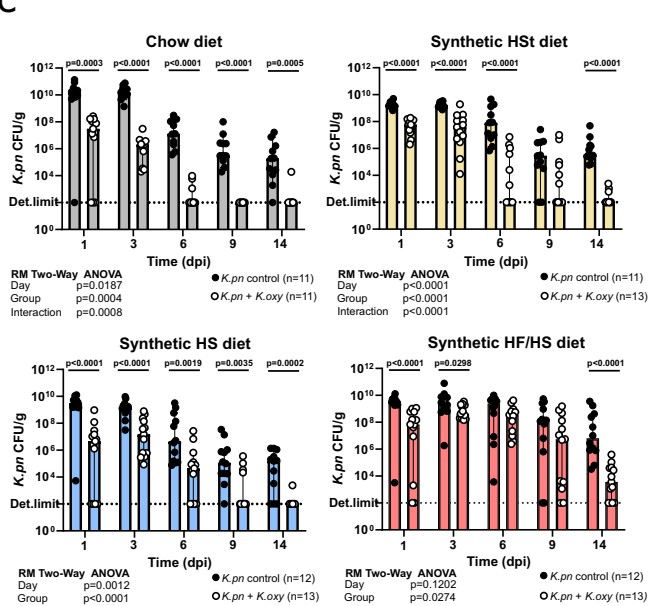

**Fig. 1 | Diet modulates *K. pneumoniae* (*K. pn*) colonization dynamics and clearance is accelerated by *K. oxytoca* (*K. oxy*) colonization. a** Schematic showing the experimental setup. Mice were placed on specific diets and ampicillin, after which one group per diet received *K. oxytoca* inoculation orally while the other group per diet received no inoculation. All mice were challenged with *K. pneumoniae* on 0 dpi. **b** Mean *K. oxytoca* CFUs with SD confirming fecal colonization on −1 dpi from three independent experiments with n = 3–6 mice/group. *P*-values represent ordinary one-way ANOVA analysis with Tukey's multiple comparison test with *\*p < 0.05, \*\*p < 0.01, \*\*\*p < 0.005, \*\*\*\*p < 0.0001.* **c** Median *K. pneumoniae* CFUs with 95% CI per group per day in mice fed standard chow, semi-synthetic, semi-synthetic high-sucrose, and semi-synthetic high-fat/high-sucrose diets from three

independent experiments with n = 3–6 mice/group. Bars represent group medians, and individual dots represent fecal samples collected from individual mice. Two-tailed *p*-values indicated at individual time points represent Mann-Whitney non-parametric rank comparison between *K. oxytoca* pre-colonized and non-colonized groups per day with *\*p < 0.05, \*\*p < 0.01, \*\*\*p < 0.005, \*\*\*\*p < 0.0001.* Global *p*-values represent a two-way repeated-measures ANOVA with Geisser-Greenhouse correction. **d** Clearance rates of *K. pneumoniae* per group per day representing the percentage of animals in each experimental group showing clearance. Number of all animals and those showing clearance are indicated in parentheses. *P*-values represent the Log-rank (Mantel-Cox) test with *\*p < 0.05, \*\*p < 0.01, \*\*\*p < 0.001, \*\*\*\*p < 0.0001.* Panel (**a**) Created with BioRender.com.

*K. oxytoca* colonization were confirmed on all dietary backgrounds despite 10-fold reduced CFU counts in HF/HS (CFU/g) (Fig. 1b). Next, all animals were challenged on 9 dpd with $5 \times 10^8$ CFU of *K. pneumoniae* MD01, a multidrug-resistant nosocomial isolate encoding a NDM-1 carbapenemase from the emerging sequence type 395. Of note, early (1-day post-infection; dpi) *K. pneumoniae* colonization levels in control mice reached similar colonization levels ($10^9$–$10^{10}$ CFU/g) on all four diets. Longitudinal quantification of *K. pneumoniae* colonization revealed significant differences between *K. oxytoca* pre-colonized mice and the control group (Fig. 1c). Initially, i.e., on 1 dpi and while the mice still received ampicillin, the inhibitory effect of *K. oxytoca* on *K. pneumoniae* colonization levels was comparable across all diets. Specifically, *K. pneumoniae* colonization was reduced 10–100 fold on 1 dpi. However, at later time points, i.e., after ampicillin supplementation was terminated, diet-specific signatures started to emerge. *K. oxytoca* pre-colonized mice on standard chow, HSt, and HS diets all showed significant reductions in pathogen colonization throughout the experiment, whereas HF/HS mice harbored largely comparable pathogen levels from 3–9 dpi, with the most pronounced CFU differences observed only at the endpoint of the experiment on 14 dpi. In line with these observations, clearance kinetics revealed significant differences in *K. pneumoniae* decolonization depending on both the host diet and the pre-colonization status of the mice (Fig. 1d). Importantly, *K. oxytoca* markedly accelerated *K. pneumoniae* decolonization regardless of the diet (Log-rank test *p*-value: < 0.0001–0.0403). Specifically, while in 0–20% of control mice, *K. pneumoniae* colonization dropped below the detection limit – referred to as clearance –, this occurred in 36–90% of *K. oxytoca* pre-colonized mice. Yet, strong diet-dependent clearance patterns were observable across the four diets. Most notably, mice on HF/HS diet began to show *K. pneumoniae* clearance only on 9 dpi and ultimately reached 36% clearance on 14 dpi, as opposed to all other diets in which clearance started three days earlier on 6 dpi and increased to 81–90% at the endpoint. The clearance differences between chow, HSt, and HS diets are less pronounced and continue to converge towards 14 dpi, when they peak between 81% (10/13 mice) on the HSt diet and 90% (10/11 mice) on the chow diet. In conclusion, these experiments revealed a general pattern of *K. oxytoca*-mediated colonization resistance in early infection across all diets and that differences in clearance kinetics following the cessation of ampicillin treatment are influenced by the host's diet.

### Sucrose utilization gene *sacX* of *K. oxytoca* does not contribute to decolonization of *K. pneumoniae* in vivo

Since two of the four tested diets contained high levels of added sucrose and considering the widespread addition of sucrose in processed foods in Westernized diets, we aimed to dissect whether direct consumption of sucrose by *K. oxytoca* contributes to the competition between the two strains. To this end, we identified through homology-based bioinformatics analysis a phosphotransferase system (PTS) in *K. oxytoca* as a putative contributor to sucrose utilization (Fig. 2a). We deleted the gene encoding the transmembrane component of the PTS, *sacX*, using our recently reported CRISPR-Cas9-based gene-editing tool[32]. The resulting deletion mutants were unable to grow in a minimal medium with sucrose as the sole carbon source, confirming its contribution to sucrose utilization in vitro (Fig. 2b). Next, we pre-colonized ampicillin-treated SPF mice (Fig. 2c) with either *K. oxytoca* WT or *K. oxytoca* Δ*sacX* (Fig. 2d), and compared *K. pneumoniae* colonization levels throughout the experiment next to a control group. Of note, the gene deletion did not decrease the competitiveness of *K. oxytoca* against *K. pneumoniae* on diets that contained no sucrose (standard chow and HSt, Fig. 2e, f). Unexpectedly, also on the two high-sucrose diets, namely HS and HF/HS, no differences were observed between *K. oxytoca* WT or Δ*sacX* pre-colonized mice (Fig. 2g, h). Similarly, this affected neither the decolonization of *K. pneumoniae* (Fig. 2e–h) nor

the colonization levels of *K. oxytoca* (Supplementary Fig. 1a–d) regardless of the sucrose content of the host diet.

Together these results show that sucrose utilization was *sacX*-dependent in vitro and it had no effect on the CFU reduction of *K. pneumoniae* by *K. oxytoca* suggesting that either other genetic features or other diet-induced differences underlie pathogen clearance in HS and HF/HS diets. Of note, the possibility that other genes might contribute to sucrose utilization in vivo in *K. oxytoca* cannot be excluded. Furthermore, since diet-dependent patterns in colonization resistance only occur after ampicillin treatment is stopped, this suggests that the recovering microbiome might drive distinct decolonization dynamics.

### Effective clearance necessitates the presence of co-colonizing gut commensals

Hence, we next aimed to investigate the direct interactions between *K. oxytoca* and *K. pneumoniae* in the absence of other gut commensals. Therefore, we first performed ex vivo competition assays in the caecum content of GF mice fed one of the four diets (Fig. 3a). After 24 hours of incubation, *K. pneumoniae* growth was significantly inhibited, 10–100-fold lower CFUs on all diets, in the presence of *K. oxytoca* compared to the monoculture control (Fig. 3b). This indicates direct competition between the two species, in line with what was observed in vivo in SPF-ampicillin mice (Fig. 1c). Next, we performed in vivo competition assays in GF mice, which allowed us to investigate the direct interspecies competition for longer periods and under physiological conditions. These co-colonization experiments were performed on standard chow and the HS diet, as representative Western-style diet with comparable clearance (Fig. 3c). Notably, the reduction of *K. pneumoniae* levels in *K. oxytoca* pre-colonized mice was transient, and independent of the diet (Fig. 3d). Specifically, as the colonization differences between the groups decreased over time, they ultimately lead to the co-existence of both species in the absence of other co-colonizing microbes (Fig. 3d, e). Thus, these data collectively indicated that although *K. oxytoca* initially impeded *K. pneumoniae* establishment in the gut, diet-dependent clearance variations likely resulted from host microbiome adaptation to specific dietary niches.

### Low-fiber synthetic diets promote distinct pathobiont-dominated microbiome signatures during post-antibiotic recovery

Thus, we assessed how the dietary background and *K. oxytoca* pre-colonization status contribute to microbiome changes that, in turn, affect pathogen clearance. To this end, we performed longitudinal amplicon 16S rRNA gene sequencing on 12 groups of mice covering the most important time points of our in vivo competition model. Firstly, mice received one of four of the aforementioned diets, and on each diet, they underwent different "treatments": (1) ampicillin treatment alone, (2) ampicillin treatment and *K. oxytoca* pre-colonization, or (3) untreated control (Fig. 4a). Of note, fecal samples for sequencing were collected before but not during ampicillin treatment (due to insufficient read counts obtained from ampicillin-treated samples) as well as at time points equivalent to sample collections after *K. pneumoniae* colonization in the co-colonization model, namely 10, 12,15,18, and 23 dpd.

First, the introduction of all semi-synthetic diets in the untreated control groups led to an overall decrease in both observed species richness and Shannon diversity scores, i.e., lower numbers of detectable amplicon sequence variants (ASVs) and less even communities, respectively, compared to mice on standard chow (Supplementary Fig. 2a–c). Fecal communities showed shifts in the microbiome composition on semi-synthetic diets, which were detectable at the earliest time points after dietary intervention, i.e., 2 dpd (Supplementary Fig. 2d, e). Linear discriminant analysis (LDA) effect size (LEfSe) analysis of samples on 2 dpd revealed a general decrease in the relative

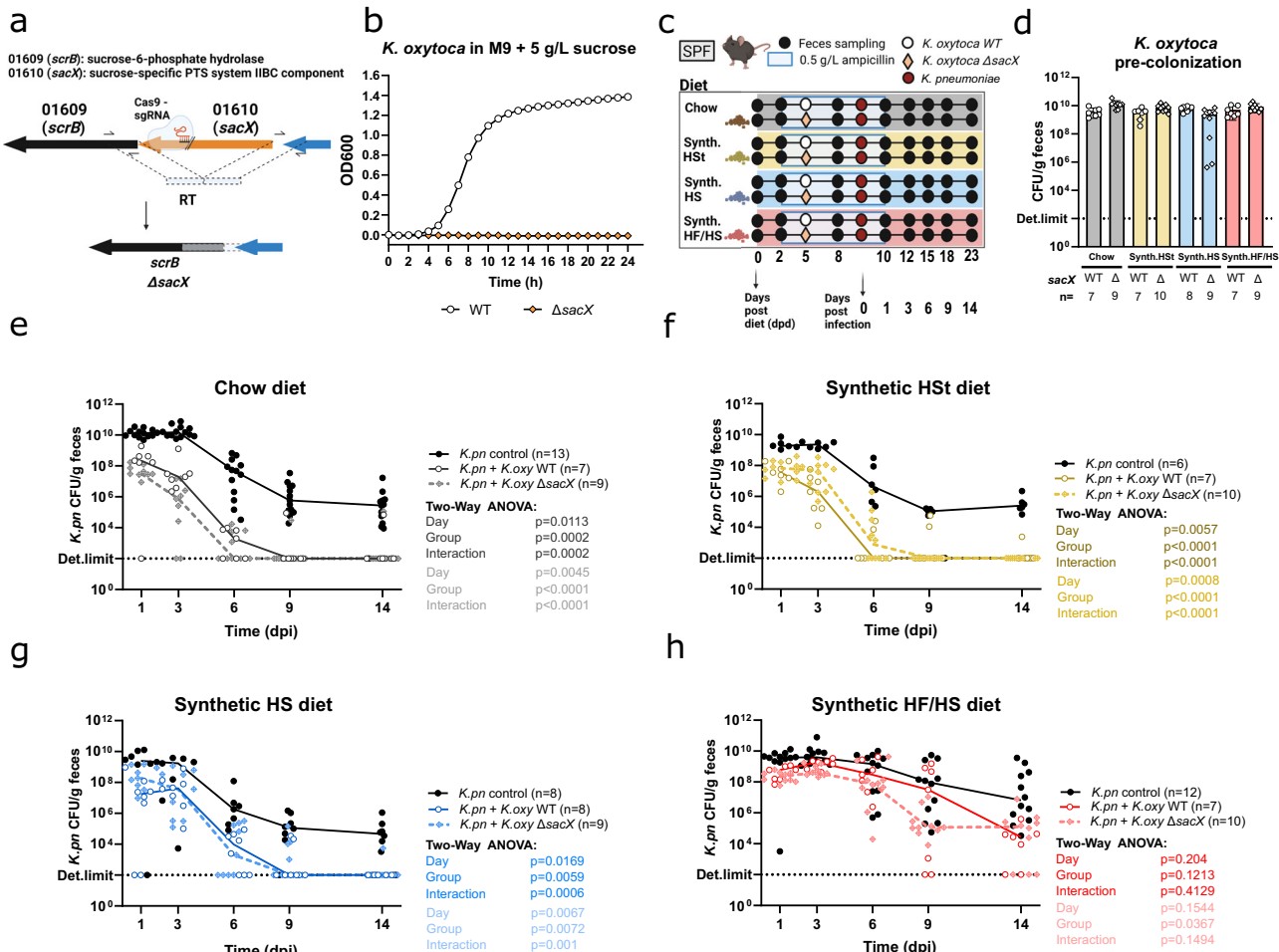

**Fig. 2 | *K. pneumoniae* CFU reduction and clearance by *K. oxytoca* is independent of sucrose utilization. a** Schematic of the genomic locus of the sucrose-specific PTS in *K. oxytoca* MK01. sacX was deleted from the genome using a CRISPR-Cas9-mediated gene editing tool. **b** *K. oxytoca* WT and ΔsacX growth on minimal medium + 5 g/L sucrose represented by longitudinal $OD_{600}$ values for 24 h, dots represent means of 3 technical replicates. **c** Schematic showing setup of animal experiment. Mice were pre-colonized with either *K. oxytoca* WT or ΔsacX strain. **d** Mean *K. oxytoca* CFUs with SD confirming fecal colonization one day before mice

were challenged with *K. pneumoniae* from 2-3 independent experiments with *n* = 3–5 mice/group. **e–h** CFUs of *K. pneumoniae* in *K. oxytoca* pre-colonized or non-colonized mice fed (**e**) standard chow, (**f**) semi-synthetic, (**g**) semi-synthetic high-sucrose or (**h**) semi-synthetic high-fat/high-sucrose diets from 2–3 independent experiments with *n* = 3–5 mice/group. Lines represent group medians. Global *p*-values represent a two-way repeated-measured ANOVA with Geisser-Greenhouse correction with *$p < 0.05$, **$p < 0.01$, ***$p < 0.005$, ****$p < 0.0001$. Panel (**a**) and (**c**) Created with BioRender.com.

abundance of Lactobacillaceae and Rikenellaceae abundance, while other families such as Acutalibacteraceae, Tannerellaceae, and Oscillospiraceae expanded in the semi-synthetic diets compared to chow diet (Supplementary Fig. 2f–h). As expected, ampicillin reduced alpha diversity, which remained significantly lower up to a week after its removal in the ampicillin-treated group (Fig. 4b). Furthermore, ampicillin treatment resulted in a drastic expansion of bacterial taxa with pathogenic potential, namely Enterococcaceae and Staphylococcaceae (Fig. 4c and Supplementary Fig. 3a, b), which may cause severe infections in vulnerable populations as a result of blooming in the gut[33,34]. Strikingly, their relative abundance was markedly lower in the presence of *K. oxytoca* (Fig. 4d). The mean relative abundance of Enterococcaceae peaked at 23, 55 and 69% on HS, HSt, and HF/HS diets, respectively, as opposed to 7% on standard chow. However, the HS diet showed a unique expansion of Staphylococcaceae at a group average of 30% in early post-ampicillin recovery (Fig. 4e–h). Of note, the expansion of Enterococcaceae was particularly prolonged up until 18 dpd on HF/HS diet with a mean relative abundance of 18%, in contrast to only 3.9% and 0.5% on HSt and HS diets, respectively. Overall, both antibiotic treatment and semi-synthetic diets exert a negative effect on gut microbiome diversity, synergizing in the expansion of

bacteria with pathogenic potential (Supplementary Fig. 4). While the host diet influences the recovery of the microbiota post-ampicillin treatment, strikingly, *K. oxytoca* pre-colonization leads to faster recovery of OTUs and global resemblance to ampicillin-naïve communities on all dietary backgrounds.

## *K. oxytoca* colonization accelerates compositional recovery of the gut microbiome

Next, we aimed to functionally profile the changes in the microbiota during post-antibiotic recovery and how they are affected by *K. oxytoca*. Specifically, we selected samples for metagenome sequencing from 6 dpi (= 15 dpd), when clearance would occur in chow, HSt, and HS diets in *K. oxytoca* pre-colonized groups, but not on HF/HS diet (Fig. 5a). Of note, for this analysis mice were not challenged with *K. pneumoniae* to focus on the direct effect of *K. oxytoca* (see also Fig. 4a). Comparative analysis of KEGG functionalities confirms higher functional similarity between untreated and *K. oxytoca* pre-colonized groups on diets supporting pathogen clearance as opposed to *K. oxytoca* pre-colonized samples on HF/HS samples, which show large variability and cluster closer to ampicillin-treated samples (Fig. 5b). Next, we aimed to compare using LEfSe analysis samples with

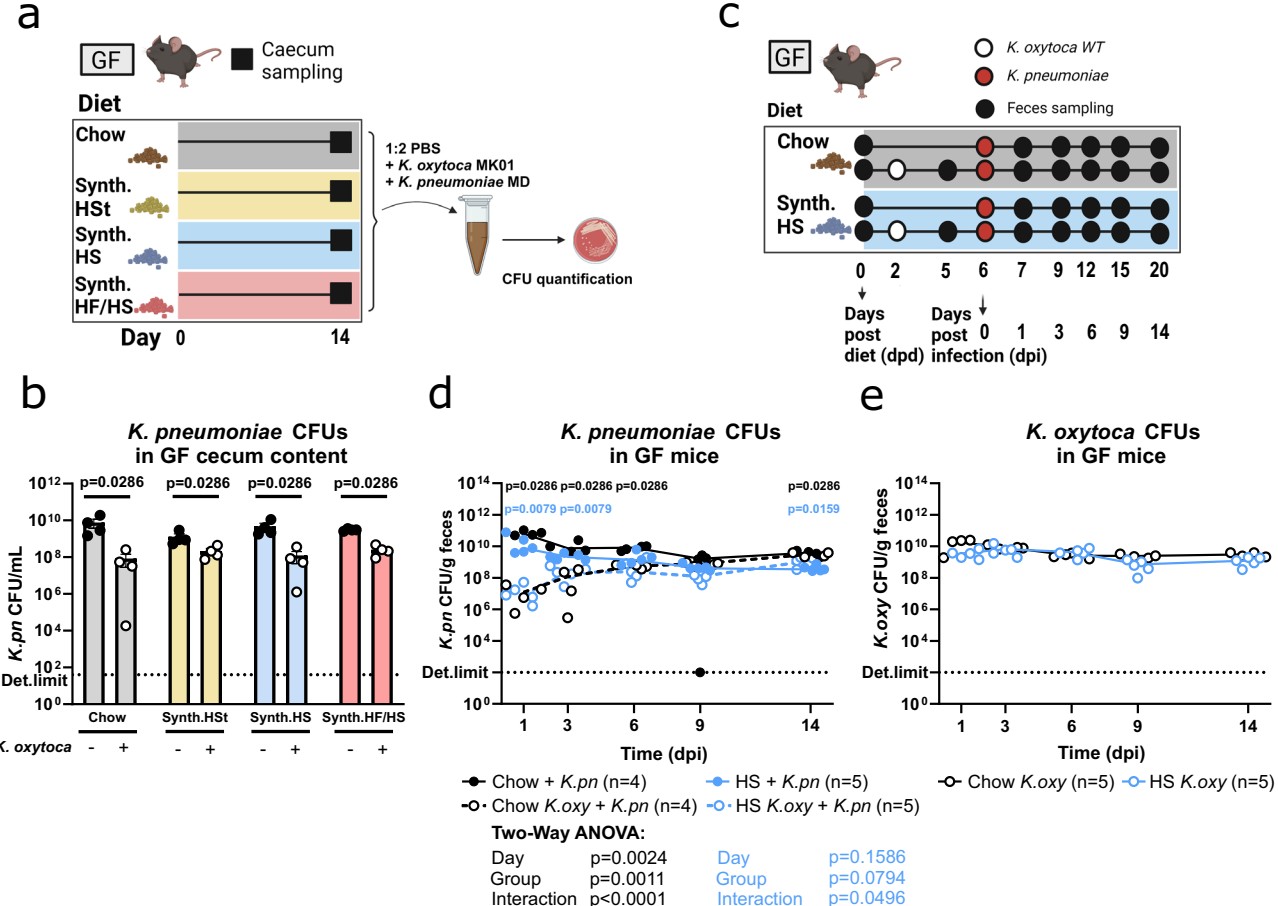

**Fig. 3 | _K. oxytoca_ directly antagonizes _K. pneumoniae_ in early infection but clearance is microbiome-dependent. a** Schematic illustrating ex vivo competition assay. Caecum content was isolated from GF mice fed one of four diets for 14 days, pooled, and used as a medium base for competition assay between _K. oxytoca_ and _K. pneumoniae_. **b** CFUs of _K. pneumoniae_ after 24 h of co-culturing with _K. oxytoca_. Dots represent the average of three technical replicates. Bars represent the mean of 4 independent experiments ($n = 4$) with SEM using different cultures of bacteria. Two-tailed $p$-values indicated represent Mann-Whitney non-parametric rank comparison with *$p < 0.05$, **$p < 0.01$, ***$p < 0.005$, ****$p < 0.0001$. **c** Schematic showing the experimental setup. One group of GF mice per diet was colonized with

_K. oxytoca_ while the other group per diet was left uncolonized. _K. pneumoniae_ challenge took place four days after _K. oxytoca_ pre-colonization, colonization levels of both bacteria were monitored for 14 days. **d, e** CFUs of (**d**) _K. pneumoniae_ and (**e**) _K. oxytoca_ from fecal samples. Lines represent group medians from one experiment with $n = 4$–5 mice/group. Global $p$-values represent a two-way repeated-measured ANOVA with Geisser-Greenhouse correction with *$p < 0.05$, **$p < 0.01$, ***$p < 0.005$, ****$p < 0.0001$. Two-tailed $p$-values indicated at individual time points represent Mann-Whitney non-parametric rank comparison between _K. oxytoca_ pre-colonized and non-colonized groups per day with *$p < 0.05$, **$p < 0.01$, ***$p < 0.005$, ****$p < 0.0001$. Panel (**a**) and (**b**) Created with BioRender.com.

clearance potential ("clearers") to those without or reduced clearance ("non-clearers") (Supplementary Data 1). The "clearer" group included the _K. oxytoca_ pre-colonized groups on chow, HSt, and HS diets, and the "non-clearers" included mice that were ampicillin-treated (all diets) or HF/HS _K. oxytoca_ pre-colonized. "Non-clearer" microbiomes are significantly enriched in _Enterococcus faecalis_, _E. gallinarum_, and _Parabacteroides goldsteinii_, while "clearers" share the enrichment of various species belonging to families such as Lachnospiraceae, Oscillospiraceae, and Rikenellaceae (Fig. 5c and Supplementary Data 1). Species enriched in "clearers" encode distinct KEGG profiles compared to those enriched in "non-clearers" (Supplementary Fig. 5a) suggesting that additional functions need to be contributed by these species to restore colonization resistance. Since nutrient competition has been previously demonstrated to contribute to _K. pneumoniae_ clearance, we next compared the functional potential to degrade different dietary- and host-derived carbohydrates, i.e., their CAZyme profiles, using dbCan. Again, species enriched in "clearer" have distinct CAZyme profiles compared to those enriched in "non-clearers" (Supplementary Fig. 5b). Moreover, "non-clearer" globally share a significant enrichment of several types of carbohydrate-active functionalities related to host-glycan degradation (mucin-type _O_-glycans, host _N_-glycans, sialic

acid, blood group B substances) (Fig. 5d) and KEGG modules related to central carbohydrate metabolism (Supplementary Fig. 6 and Supplementary Data 3), while "clearers" show enrichment in complex-carbohydrate-related functionalities targeting e.g., peptidoglycans, glycogen, starch, and arabinan (Fig. 5d) and KEGG modules related to drug resistance and aromatics degradation (Supplementary Fig. 6 and Supplementary Data 3). Thus, clearance-promoting samples are associated with the emergence of an array of bacterial species from multiple families with plant- rather than host-derived catabolic capacity. This shows that _K. oxytoca_ promotes, in a diet-dependent manner, the recovery of a metabolically diverse repertoire of functionally beneficial commensals.

**Rapid commensal outgrowth post-ampicillin treatment is associated to antibiotic degradation by _K. oxytoca_**
Following the cessation of antibiotic supplementation, residual antibiotics in the intestinal content may impede the growth of commensal organisms for an extended period due to the gradual excretion and degradation of the antibiotics[35]. Since _K. oxytoca_ is resistant to ampicillin by producing β-lactamases, we hypothesized that microbiome recovery is indirectly facilitated by the β-lactamase production of _K._

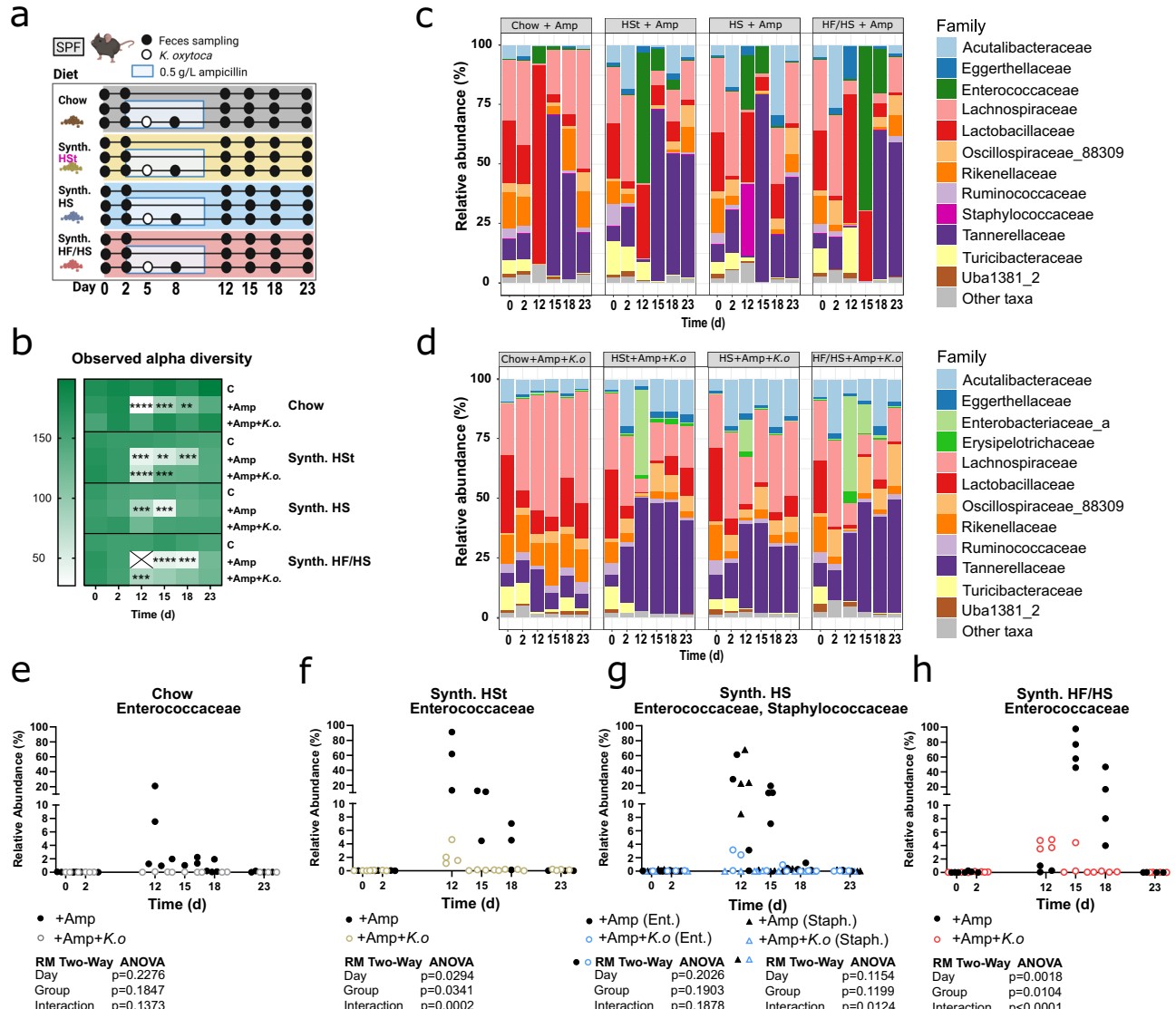

**Fig. 4 | K. oxytoca colonization facilitates α-diversity recovery and prevents pathobiont outgrowth. a** Schematic showing the experimental setup. All mice were switched to one of four diets on day 0. Two days later, two groups on each diet began ampicillin treatment while one group per diet was left untreated. On day 5, one ampicillin-treated group was colonized with *K. oxytoca* while the other ampicillin-treated group received no bacterial inoculation. Fecal samples were collected on the indicated days. **b** Heatmap depicting fecal microbiota α-diversity represented by the number of observed OTUs from one experiment with n = 3–4 mice/group. *P*-values represent multiple unpaired *t* tests with the Holm-Sidak method with *p < 0.05, **p < 0.01, ***p < 0.005, ****p < 0.0001 comparing the ampicillin-treated groups to the untreated control on each diet. Further details about SE of difference, t ratio, degree of freedom, and adjusted *p*-values can be found in the Source Data file. **c, d** Relative abundances of the 12 most abundant families shown as group average for mice treated with ampicillin (**c**) without and (**d**) with subsequent *K. oxytoca* colonization. **e–h** Longitudinal changes in the relative abundances of families (**e, g**) Enterococcaceae and (**f, h**) Staphylococcaceae (**e, f**) without and (**g, h**) with *K. oxytoca* pre-colonization. Colors represent different diets, dots represent individual mice. Panel (**a**) Created with BioRender.com.

*oxytoca*. To test this, we sampled ampicillin-treated mice either with or without prior *K. oxytoca* pre-colonization 10 dpd (= 1 dpi), and isolated their caecum contents (Fig. 6a, b). Firstly, a colorimetric test of β-lactamase activity showed measurable degradation of the β-lactam nitrocefin in the sterile-filtered caecum content of *K. oxytoca* pre-colonized mice on all dietary backgrounds (Supplementary Fig. 7). Next, to assess the inhibitory potential of ampicillin residues in the gut, sterile-filtered caecum content supernatant was supplemented with rich media supporting the growth of the ampicillin-sensitive murine gut commensal *Limosilactobacillus reuteri* (strain I48)[36] (Fig. 6c and Supplementary Fig. 8b). Cultures supplemented with ampicillin-treated intestinal content showed no measurable growth after 48 h, indicating strong growth inhibition of *L. reuteri* (Fig. 6d). On the contrary, *K. oxytoca* pre-colonized intestinal contents were growth-permissive to *L. reuteri*, although revealing differences between the

diets. Interestingly, growth curves recorded over 48 h showed highly variable final optical density, with standard chow diet supporting the highest growth to around $OD_{600} = 1.4$, while in contrast, HF/HS content showed a significantly lower final $OD_{600}$ of 0.4 in the stationary phase using Dunn's multiple comparisons (Fig. 6d). Serial dilution of the supplemented caecum content supernatant provided additional quantification of the extent of growth inhibition on *L. reuteri*. While dilution of *K. oxytoca* pre-colonized contents showed no further increase of the already robust growth, ampicillin-treated contents started supporting the growth of *L. reuteri* at 64-128x dilution (Fig. 6e). To further support the conclusion that antibiotic residues and not differences in nutrients were responsible for the growth inhibition, the commonly used, ampicillin-sensitive laboratory strain of *E. coli* MG1655 was transformed with a plasmid carrying a β-lactamase gene (pGEX) to induce ampicillin resistance (Supplementary Fig. 8c, d). Then, WT and

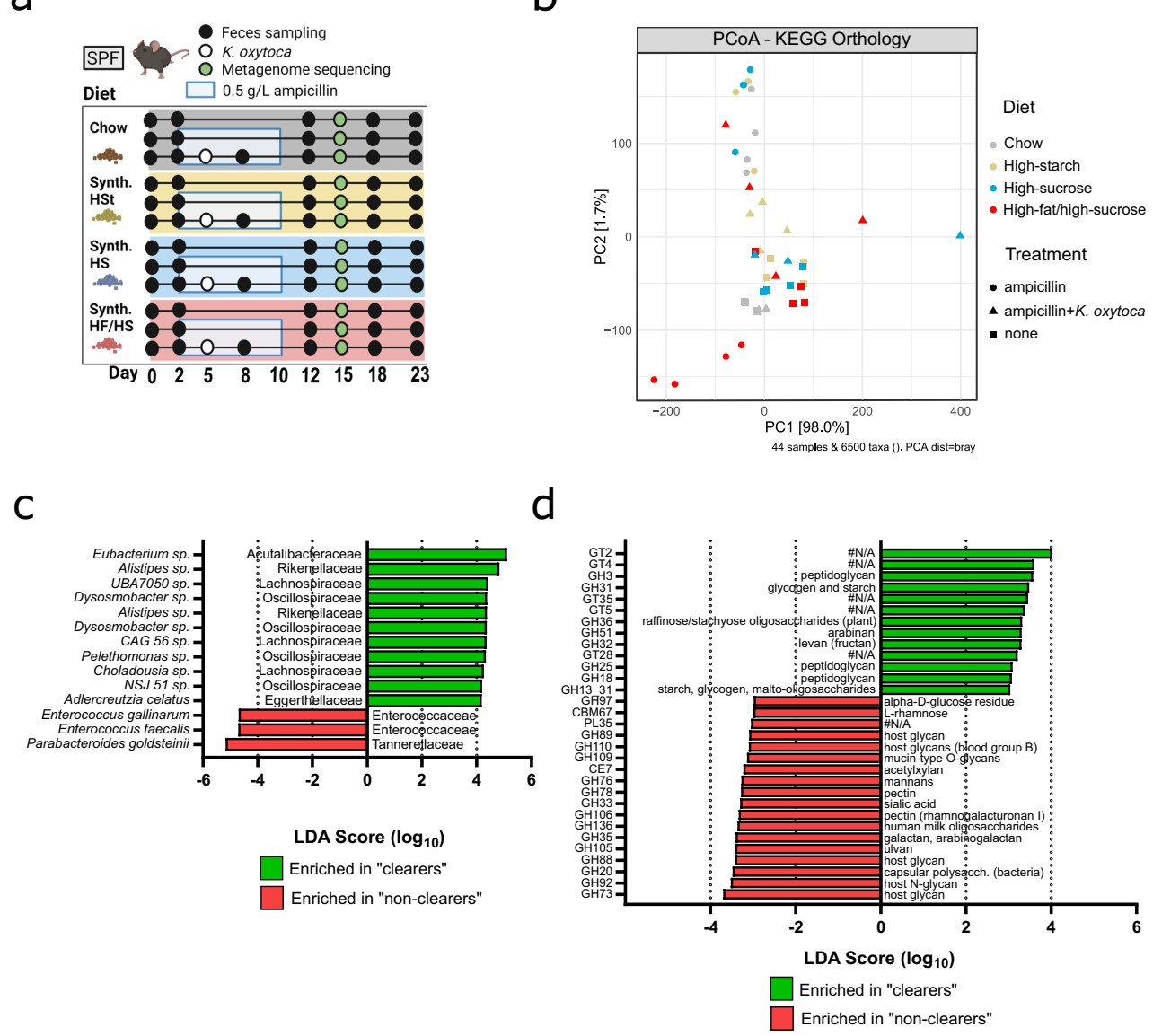

**Fig. 5 | Metagenome sequencing reveals diet-specific delays in microbiome recovery in mice fed HF/HS diet on day 15. a** Schematic showing the experimental setup. All mice were switched to one of four diets on day 0. Two days later, two groups on each diet began ampicillin treatment while one group per diet was left untreated. On day 5, one ampicillin-treated group was colonized with *K. oxytoca* while the other ampicillin-treated group received no bacterial inoculation. Fecal samples were collected for metagenome sequencing on 15 dpd. **b** PCoA plot calculated by Bray-Curtis distances of annotated KEGG functional modules in fecal samples collected on day 15 from mice either untreated, ampicillin treated or ampicillin treated and *K. oxytoca* colonized from one experiment with *n* = 3–4 mice/ group. **c**, **d** Analysis of differentially abundant (**c**) bacterial taxa (from de novo generated MAG profiles) and (**d**) carbohydrate-active enzymes (CAZymes) in mice that show *K. pneumoniae* clearance versus mice that show no clearance by LEfSe. Panel (**a**) Created with BioRender.com.

plasmid-bearing clones were grown in conditions similar to *L. reuterii* I48. Supplementation of ampicillin-treated caecum content fully inhibited WT *E. coli* cells similarly to *L. reuteri* I48, while *E. coli*-pGEX clones grew without inhibition in the same experimental setup (Fig. 6f). Finally, we quantified ampicillin present in the caecum content using LC-MS measurements. Our data showed that ampicillin is detectable at 100–1000-fold higher levels in the caecum content of animals not pre-colonized by *K. oxytoca* (Fig. 6g and Supplementary Data 2). Furthermore, among *K. oxytoca* pre-colonized mice, we measured the highest ampicillin levels in HF/HS mice (AUC = 32 on HF/HS diet vs. AUC = 2.3 on chow), which subsequently showed the lowest *K. pneumoniae* clearance. Thus, we identified that residual ampicillin levels are lower in the guts of *K. oxytoca* pre-colonized mice, and showed detectable β-lactamase activity in isolated caecum content. Lower ampicillin burden on the commensal microbes and a

nutritionally diverse gut environment are linked to a faster recovery of the gut microbiome, which ultimately improves the clearance of pathogens from the gut.

## Discussion

In recent years, numerous studies focused on the nutrient competition between gut microbes, especially between commensals and pathogens[9–11,15]. Even though several specific carbohydrates such as β-glucosides[9], galactitol[10,11], fructose[15], and gluconate[16] have been identified through various approaches as important contributors to pathogen decolonization, the host's dietary context is often overlooked in these studies. Consequently, these observations might be context-dependent, given the fundamental differences in the ecological and dietary landscape of the gut shaped by different dietary habits[37]. The diet of populations in industrial countries – often

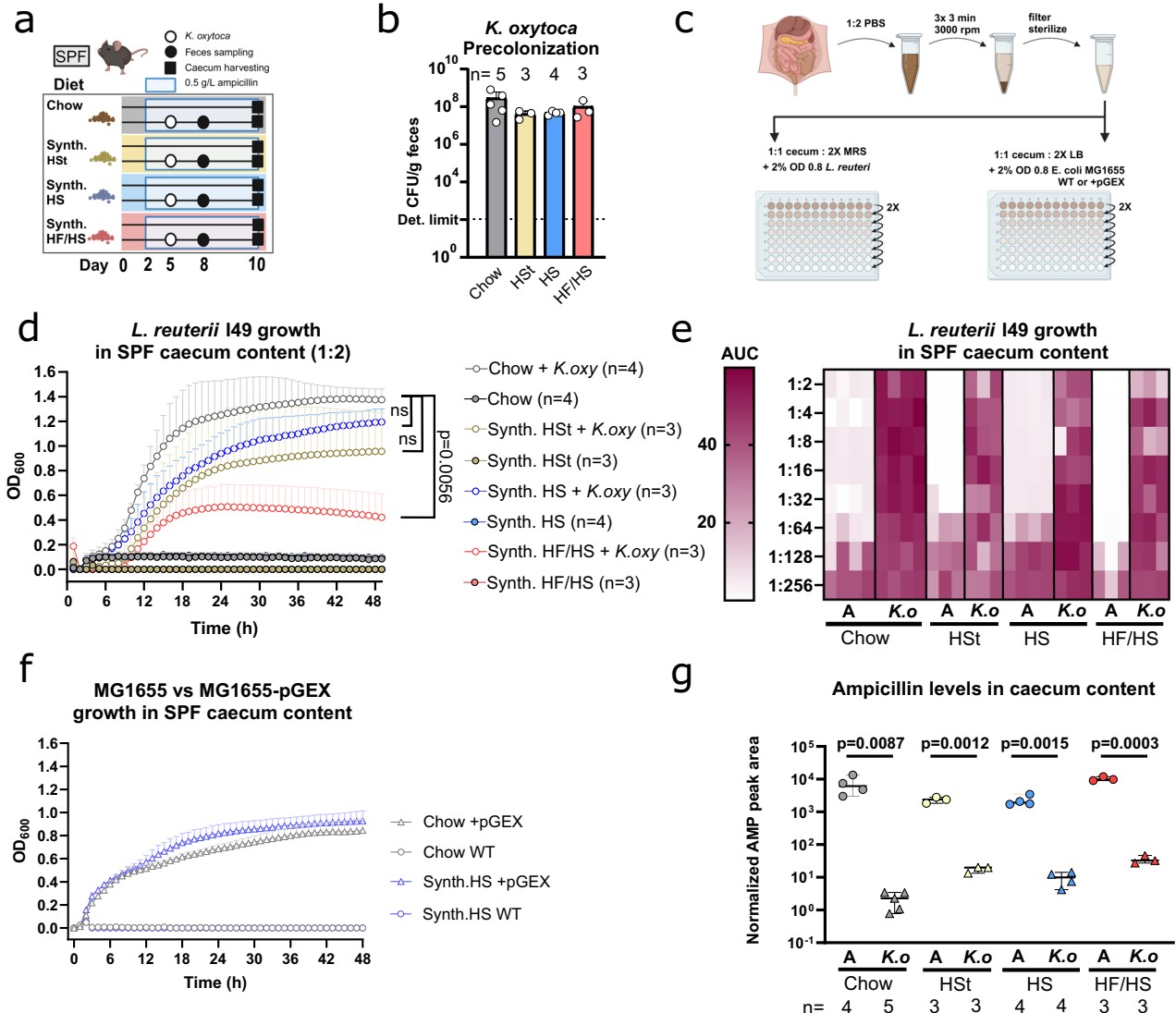

**Fig. 6 | Ampicillin degradation by *K. oxytoca* facilitates *Limosilactobacillus reuteri* outgrowth in a diet-dependent manner. a** Schematic showing in vivo experimental setup. **b** Fecal colonization was confirmed by CFU quantification on day 8. **c** Schematic showing ex vivo growth assays. In brief, isolated caecum contents (*n* = 3–4 mice/group) were diluted with 2X volume of PBS, after which they were centrifuged in three consecutive steps, removing the supernatant each time. Final supernatants were filtered using a 0.22 μm filter and serially diluted with PBS. 2X MRS and 2X LB were supplemented to support the growth of *L. reuteri* I49 and *E. coli* MG1655, respectively. **d** *L. reuteri* I49 growth in a mixture of (ratio 1:1) caecum content supernatants and MRS (final concentration: 1x) for 48 h. Displayed growth curves represent the mean of three-four biological replicates with standard deviation. *P*-values on the right side represent Kruskal Wallis test with Dunn's multiple comparisons of the biological replicates in each condition at 48 h with

$*p < 0.05$, $**p < 0.01$, $***p < 0.005$, $****p < 0.0001$ (**e**) Heatmap depicting *L. reuteri* I49 growth represented by area under the curve (AUC) measurements after 48 h in a mixture of (ratio 1:1) caecum content supernatants and MRS (final concentration: 1x). **f** *E. coli* MG1655 growth in a mixture of caecum content supernatants and LB (final concentration 1:1) for 48 h. Displayed growth curves represent three biological replicates with standard deviation. **g** Weight-normalized AMP peak areas determined in cecal extracts of mice (*n* = 3–4 mice/group) treated with ampicillin and mice treated with ampicillin and colonized with *K. oxytoca* fed different diets. Data are displayed as individual biological replicates (mean of double injection), and bars indicate the median with 95% confidence intervals. The indicated two-tailed *p*-values represent unpaired *t* test with $*p < 0.05$, $**p < 0.01$, $***p < 0.005$, $****p < 0.0001$. Panel (**a**) and (**c**) Created with BioRender.com.

referred to as the Westernized diet – has been the particular focus of microbiome-related studies. Still, only a few discuss it in the context of colonization resistance[38,39]. Thus, we aimed to investigate how the interspecies competition between *Klebsiella* species is affected under different Westernized diets. Our data demonstrates that *K. pneumoniae* growth reduction and, ultimately clearance were significantly accelerated in the presence of *K. oxytoca* compared to the control group on all diets. Notably, the decolonization kinetics of the pathogen were significantly affected by diet. Firstly, *K. pneumoniae* colonization remained elevated on the HF/HS diet even after the cessation of the ampicillin treatment, after which both *K. pneumoniae* and *K. oxytoca* colonization were reduced on all other dietary

backgrounds. Even though *K. oxytoca* colonization was comparable across the different diets, *K. pneumoniae* specifically showed increased persistence on the HF/HS diet, leading to markedly lower clearance rates. As most dietary fats and simple sugars are absorbed in the small intestine, they provide little nutrient benefit to gut commensals in the colon, which primarily rely on complex fibers for nutrient acquisition[40]. This often leads to low diversity of colonic microbiomes, resulting in weaker resilience to invading pathogens[41]. Other studies also support these observations, as Enterobacteriaceae expansions on HF/HS diet have been previously described[42,43]. Even though we show that *K. oxytoca* antagonizes *K. pneumoniae* in early colonization phases, pathogen clearance does not occur without the

recovering gut microbiome, which we hypothesized would explain clearance differences between dietary backgrounds.

Microbiota analysis revealed that alpha diversity metrics decreased after dietary intervention on all three semi-synthetic diets, and notable compositional changes appeared on day 2. Furthermore, the analysis showed vast differences in the relative abundances of bacteria during the post-antibiotic recovery period between ampicillin-treated mice with and without *K. oxytoca* colonization and under different dietary conditions. For example, Enterococcaceae showed expansion in chow and HSt groups on day 12 and HF/HS group on day 15, and the expansion of the family Staphylococcaceae was unique to mice on the HS diet. Alpha diversity measures consistently showed accelerated recovery due to *K. oxytoca* colonization. Ampicillin-treated mice exhibited significantly lower diversity and Shannon scores than untreated mice from days 15–18 across all dietary backgrounds. However, the presence of *K. oxytoca* reduced these discrepancies in all dietary backgrounds. Notably, the previously detected expansions in the relative abundance of Enterococcaceae and Staphylococcaceae were not observed in *K. oxytoca* colonized mice. Furthermore, our data confirms previous studies that show how post-antibiotic microbiome recovery is context-dependent and is largely shaped by the host diet[41]. In a study by Ng et al., conventional mice consuming a fiber-deficient diet showed lower overall alpha diversity and drastically decreased density of anaerobic bacteria in fecal samples following ciprofloxacin treatment[41]. It was concluded that the severe loss of anaerobic taxa can lead to invasion by opportunistic pathogens. While we did not quantify absolute bacterial densities throughout our study, which is a limitation, these results align with our observations regarding the decreased alpha diversity and the slower recovery of the gut microbiome post-ampicillin treatment. Both the Enterococcaceae and Staphylococcaceae expansion, as well as the persistence of high *K. pneumoniae* CFUs can be the result of uncontested niches present in fiber-deprived conditions.

Ex vivo assessment of intestinal content collected from ampicillin-treated mice with or without *K. oxytoca* pre-colonization showed differences in the residual ampicillin amounts in the gut, which we attributed to β-lactamase activity by *K. oxytoca*. Furthermore, we validated that differential ampicillin quantities in the gut have meaningful physiological consequences on the host, as it enables faster outgrowth of ampicillin-sensitive gut commensals, such as *L. reuteri* in our model system. This was an important observation, as it provides a framework for understanding how vastly different dietary backgrounds can support *K. oxytoca*-driven pathogen clearance such as standard chow and HS diets. However, the importance of a diet should not be understated, especially concerning the presence of dietary fiber. Our data consistently shows the highest microbiome diversity, pathogen clearance, and fastest microbiome recovery on the chow diet. As antibiotics deplete resident bacteria fermenting dietary fibers, the levels of SCFAs drop in the gut, depriving colonocytes of a vital energy source[5]. Consequently, bacterial communities in the gut become less diverse and offer weaker protection against pathogen colonization via nutrient competition[44].

It is important to note, that *K. oxytoca* has been associated with antibiotic-induced hemorrhagic colitis in antibiotic-treated patients who tested negative for *Clostridioides difficile*[45,46]. Studies found that a causative agent in the pathogenesis of colitis is pyrrolobenzodiazepine tilivalline[47], which (alongside closely related compound tilimycin) was recently shown to exert antimicrobial activity against Salmonella Typhimurium, providing CR to ampicillin-treated conventional mice[48]. These contradictory outcomes highlight the need for vigorous safety testing of all bacterial strains with biotherapeutic potential as well as acknowledging the context-dependent nature of microbial strategies of gut microbes.

In summary, we suggest that short-term direct competition between *K. oxytoca* MK01 and *K. pneumoniae* in early colonization and the MK01-mediated degradation of antibiotic residues in the gut synergize in restoring colonization resistance on different dietary backgrounds. We provide evidence that both components are required for efficient pathogen clearance, as past studies from our group showed that not all ampicillin-resistant bacteria would enable pathogen decolonization[9]. Thus, complementary experiments show that neither the metabolic activity of *K. oxytoca* MK01 nor the ability to degrade ampicillin is sufficient to promote pathogen clearance. In line with past studies, a complex community of gut microbes must be present to maintain and preserve pathogen-free status in the gut. Our results demonstrate the significant contribution of the host diet to colonization resistance against human pathogens, and they provide insights into the beneficial effects of antibiotic degradation in the distal GI tract on the gut microbiome.

## Methods

### Ethics statement
All animal experiments were performed in agreement with the guidelines of the Helmholtz-Zentrum für Infektionsforschung, Braunschweig, Germany, the national animal protection law (Tierschutzgesetz (TierSchG)), the animal experiment regulations (Tierschutz- Versuchstierverordnung (TierSchVersV)), and the recommendations of the Federation of European Laboratory Animal Science Association (FELASA). This study was approved by the Lower Saxony State Office for Nature, Environment and Consumer Protection (LAVES), Oldenburg, Lower Saxony, Germany; permit No. 33.19-42502-04-21/3818 and 33.33-42502-04-21/3795.

### Mice
C57BL/6 N SPF-H mice were bred and maintained at the in-house animal facility of the Helmholtz Center for Infection Research (HZI) under specific pathogen-free (SPF) conditions. Germ-free (GF) C57BL/6NTac mice were bred in isolators in the GF facility at the HZI. Mice were kept under a 12 h light cycle (lights on from 7.00 am – 7.00 pm), Temperature 22 °C +/− 1, Humidity: 55% +/− 10, and housed in groups of 3–6 mice per cage. Sterilized food and water were available to all animals *ad libitum*.

The experimental animals were all age- and sex-matched, between the ages of 8 and 16 weeks. The number and sex of all experimental animals are indicated in Supplementary Data 4. GF mice were kept in airtight, individually ventilated ISOcages to prevent contamination. All mice were euthanized by asphyxiation with $CO_2$ and cervical dislocation.

### Bacterial strains
*K. oxytoca* MK01 was isolated from a human donor and described in detail in a previous study[9]. The strain is available upon completion of an MTA from the corresponding author. *K. pneumoniae* MD01 is an OXA-48-positive carbapenem-resistant patient isolate (sequence type 395) obtained from the University Hospital Magdeburg. It was previously used to study intestinal colonization and colonization resistance[9]. The strain is available upon request from Prof. Dirk Schlüter (Schlueter.Dirk@mh-hannover.de).

### Diets
Animals received a standard chow diet sterilized by irradiation available at the animal facility of the HZI (Ssniff Spezialdiäten: V1534). In specific experimental set-ups, mice received one of three semi-synthetic diets: (1) 10 kJ% fat, 1% sucrose (Ssniff Spezialdiäten: E157452-047), (2) 10 kJ% fat, 16% sucrose (Ssniff Spezialdiäten: E157451-047) or (3) 45 kJ% fat (Lard), 21.1% sucrose (Ssniff Spezialdiäten: E15744-347). Further details of the nutritional composition of each diet are summarized in Supplementary Data 5. All diets were ordered sterilized by γ-irradiation. Diets were available to all mice *ad libitum*.

### In vivo colonization of *Klebsiella* strains

SPF mice were treated with 0.5 g/L ampicillin in their drinking water for three consecutive days to disrupt colonization resistance of the resident microbiota before pre-colonization with *Klebsiella oxytoca* (*K. oxytoca* MK01[9], unless otherwise indicated). GF animals did not receive any antibiotic treatment because of the lack of a resident microbiota. In in vivo competition experiments, mice were challenged with *K. pneumoniae* MD01 four days after stable *K. oxytoca* colonization was established. All bacterial cultures were prepared in the same manner. Briefly, overnight bacterial cultures were back-diluted 1:50 in 25 ml fresh liquid LB-medium and were grown for 4 h at 37 °C while shaking continuously at 80 rpm. Cultures were centrifuged for 15 min at $500 \times g$, then the supernatant was removed, and pellets were washed twice in PBS. Mice were gavaged orally with $5 \times 10^8$ CFUs of *Klebsiella* cells diluted in 200 μl PBS and rectally with $2.5 \times 10^8$ CFUs of *Klebsiella* cells diluted in 100 μl PBS to facilitate stable and efficient *K. oxytoca* colonization.

### In vivo quantification of *Klebsiella* strains

*Klebsiella* colonization levels were determined from fresh fecal samples collected from each mouse. Fecal pellets were weighed, then circa 100 μl of 0.5 mm zirconia/silica beads were added with 1 ml of sterile PBS. Samples were then homogenized with the BioSpec Mini-Beadbeater-96 once for 50 s at 2400 rpm. Next, 200 μl of homogenate was removed from the tubes and serially diluted to $10^7$. 25 μl of all eight serial dilutions were plated on agar plates. For determining *K. oxytoca* colonization levels, MacConkey agar plates were used. *K. oxytoca* and *K. pneumoniae* colonies were differentiated by unique morphological features. Following *K. pneumoniae* challenge, dilutions were plated on both LB agar plates supplemented with 50 μg/ml kanamycin for *K. pneumoniae* selection and quantification and MacConkey agar plates for *K. oxytoca* CFU quantification.

### Gene editing in *K. oxytoca* MK01

Deletion mutants of *sacX* were generated using our recently published, optimized Cas9-mediated gene editing tool[32]. In brief, overnight cultures of *K. oxytoca* were back-diluted 50-fold and grown until an $OD_{600}$ of 0.6–0.8 in LB medium. Electrocompetent cells were generated via three rounds of washing the bacterial pellet with ice-cold 10% glycerol solution and centrifuging for 5 min at $7.200 \times g$, after which 50 μl aliquots were stored at −80 °C. Competent cells were transformed with 50 ng of pCas-apr plasmid. Transformants were selected on LB agar plates supplemented with 60 μg/ml apramycin. Next, a single colony of *K. oxytoca*-pCas-apr was cultured overnight at 30 °C in an LB medium supplemented with apramycin. After 50-fold back-dilution, the culture was grown at 30 °C until an $OD_{600}$ of 0.15–0.2, when 10 v/v % of 20% LB-L-arabinose was added for lambda Red induction. The culture was grown for 2 h at 30 °C, after which they were prepared electrocompetent as detailed above. In the second round of transformation, 200 ng of *sacX*-specific gRNA encoding pSG-spec was co-transformed with a linear repair template assembled via overlap extension PCR from 500–500 bp upstream-downstream homology arms of *sacX*. Transformants were selected on LB agar plates supplemented with 60 μg/ml apramycin and 300 μg/ml spectinomycin. Deletion of *sacX* was confirmed with colony PCRs using primers targeting the genome outside of the edited region. Deletion mutants were cured of both plasmids before further downstream experiments.

### In vitro verification of sucrose utilization of *K. oxytoca*

For assessment of sucrose utilization of *K. oxytoca* WT and Δ*sacX*, strains were cultured overnight in liquid LB medium, cultures were streaked out on R2A agar plates and incubated at 37 °C. After 16 h, bacterial lawns were scraped into PBS and adjusted to $OD_{600}$ of 0.2, then inoculated in a 1:20 ratio into M9 media with 5 g/L sucrose in a 96-well format under aerobic incubation with continuous shaking. $OD_{600}$

was recorded every hour using a microplate reader by Biotek (Logphase600). Normalized values were visualized using the software GraphPad Prism (version 8.2.1).

### Ex vivo competition assay between *K. pneumoniae* and *K. oxytoca*

For the competition assays between *K. oxytoca* and *K. pneumoniae*, we collected caecum contents from GF mice fed one of four different diets for two weeks. Isolated caecum contents were pooled from 3–6 individual mice, then diluted 1:2 with sterile PBS and stored at − 20 °C until further use. This pooled cecal content was used in independent experiments with different cultures of bacteria to assess competition. *K. oxytoca* and *K. pneumoniae* were grown overnight in LB medium, then were diluted to $OD_{600}$ of 1 and $OD_{600}$ of 0.2, respectively. 20 μl of *K. oxytoca* and 10 μl of *K. pneumoniae* were co-inoculated into 225 μl of PBS-diluted germ-free caecum content in a 96-well format. After 24 h, assays were plated in serial dilutions on MacConkey agar plates for *K. oxytoca*. The next day *K. oxytoca* and *K. pneumoniae* colonies were quantified based on distinguishable colony morphologies.

### Fecal microbial community DNA extraction

Fecal samples were collected and stored at − 80 °C until further use. DNA extraction was performed using a previously described[49] phenol-chloroform-based protocol. In brief, each fecal sample was suspended in 500 μl Buffer A, 200 μl 20% SDS, and 500 μl phenol:chloroform:isoamyl alcohol (24:24:1) with 100 μl of 0.1 mm zirconia/silica beads. Samples were then homogenized with the BioSpec Mini-Beadbeater-96 three times for 5 min at 2400 rpm; samples were kept on ice in between for 5 min to prevent overheating of the samples. Next, homogenized samples were centrifuged at $8000 \times g$ for 3 minutes at 4 °C. The aqueous phase of the supernatant was then transferred to a new microcentrifuge tube and an equal amount of phenol:chloroform:isoamyl alcohol was added and mixed thoroughly. Samples were centrifuged at maximum speed for 5 min. Once again, the aqueous phase of the supernatant was transferred to a new microcentrifuge tube and mixed thoroughly with 60 μl 3 M sodium-acetate and 600 μl of cold 100% isopropanol. Samples were incubated at − 20 °C overnight or for at least an hour. Subsequently, the samples were centrifuged at maximum speed for 30 min at 4 °C, after which the supernatant was removed. The remaining pellets were washed with 1 ml of 70% ice-cold ethanol and centrifuged for 15 min. The supernatant was carefully removed, and any residual ethanol was removed by a vacuum centrifuge. The dried pellets were re-suspended in 100 μl TE buffer and 1 μl RNase A (100 mg/mL) and incubated at 50 °C for 30 min. Finally, samples were purified from RNase A and PCR inhibitors using a Spin column PCR purification kit by BioBasic following to the manufacturer's protocol. Purified DNA samples were stored at − 20 °C until further use.

### 16S rRNA gene amplification and sequencing

Amplification of the 16S rRNA gene V4 hypervariable region (F515: GTGCCAGCMGCCGCGGTAA, R806: GGACTACHVGGGTWTCTAAT) was performed based on an established protocol previously described[50]. In brief, input DNA for sequencing PCR was normalized to 25 ng/μl and performed with unique 12-base Golary barcodes incorporated via specific primers (Sigma). After pooling and normalization to 10 nM, PCR amplicons were sequenced on an Illumina MiSeq platform via 250 bp paired-end sequencing (PE250). The resulting raw reads were demultiplexed by idmp (https://github.com/yhmu/idemp) according to the given barcodes[50]. Libraries were processed, including merging the paired-end reads, filtering the low-quality sequences, dereplication to find unique sequences, singleton removal, denoising, and chimera checking using the USEARCH pipeline 11.0.667[51]. In brief, reads merged by fastq_mergepairs command (parameters: maxdiffs 30, pctid 70, minmergelen 200, maxmergelen 400), filtered for low

quality with fastq_filter (maxee 1) and singletons using fastx_uniques command (minuniquesize 2). To predict biological sequences (ASVs, zOTUs) and filter chimeras, we used the unoise3 command (minsize 10, unoise_alpha2), following the amplicon quantification using usearch_global command (strand plus, id 0.97, maxaccepts 10, top_hit_only, maxrejects 250). Taxonomic assignment was conducted by Constax[52] (classifiers: rdp, sintax, blast) using the GreenGenes2 database[53] and summarizing into biom-file for visualization in phyloseq[54] and downstream analysis. Low-abundance OTUs with less than 0.5% abundance in all samples were excluded.

### Shotgun metagenome sequencing

Metagenomic libraries were prepared using the Illumina DNA PCR-Free Library Kit and the IDT for Illumina DNA/RNA UD Indexes for previously isolated community DNA. The libraries were prepared following the manufacturer's instructions. Quantification of library concentrations was performed using Qubit ssDNA Assay Kit followed by further quantification with KAPA Library Quantification Kit for Illumina. Sequencing was performed on the NovaSeq S4 PE150 platform targeting a depth of 25 million reads per sample. The sequencing output was analyzed using a mouse gut metagenome catalog (iMGMC) developed in our group[55]. In brief, reads were mapped to a catalog of 1296 metagenome-assembled genomes (MAGs) for taxonomic annotation, while with bwa-mem2[56], a collection of 4.6 million unique genes was used for functional annotation[55]. Furthermore, reads were assembled via megahit[57] and binning via Metawrap[58]. The resulting bins were filtered for 200 MAGs (completeness > 50%, contamination < 10%) via CheckM[59]. MAGs were generally annotated via Bakta[60], Kegg KO profiles via KofamScan[61], and CAZymes were annotated using the dbCAN3 tool[62] and used to create additional abundance profiles from metagenomics samples via bwa-mem2 (Supplementary Fig. 5).

### Ex vivo assessment of β-lactamase activity

Ampicillin-treated mice with or without prior *K. oxytoca* colonization were sacrificed, caecum content was isolated from each individual mouse ($n = 3$–$4$) and diluted in PBS 1:2. Then, caecum content solutions were centrifuged 3 times at $3000 \times g$ for 5 min, with careful transfer of the supernatant after each step. Finally, supernatants were filter sterilized using 0.2 µm filters. Nitrocefin, a chromogenic β-lactam (Sigma-Aldrich, 484400) was stored in DMSO in a 20 mM solution at $-20\,°C$. The working solution was prepared fresh, shortly before use by dilution with PBS to 1 mM. Finally, 10 v/v% nitrocefin was added to sterilized caecum contents, light absorption was recorded at 486 nm.

### Ex vivo outgrowth assay with *L. reuteri* I49 and *E. coli* MG1655 and subsequent data analysis

Caecum contents were isolated from individual mice as described in the previous section. Next, sterilized and serially diluted caecum content supernatants (with PBS from 1:2 to 1:256) were supplemented 1:1 with 2X MRS and 2X LB liquid media to support the growth of *L. reuteri* I49 and *E. coli* MG1655, respectively. 2% of $OD_{600} = 0.8$ *L. reuteri* I49 or *E. coli* MG1655 culture was inoculated and incubated under aerobic conditions for 48 hours with continuous shaking. $OD_{600}$ values were recorded hourly for 3–4 biological replicates (representing individual mice the caecum content was obtained from) and three technical replicates. The mean of the three technical replicates was retrieved and normalized to the mean of the first two measured values due to the highly variable optical density of the different caecum content batches. AUC calculations for *L. reuteri* I49 growth were performed based on the 3–4 biological replicates. Since the data was already normalized in previous steps a baseline of $Y = 0$ was applied.

### LC-MS/MS Sample Preparation

Cecal samples (ca. 100 mg) were thawed on ice. 2 ml homogenizing tubes were pre-filled with 1 ml cold extraction solvent (0.1 µg/ml caffeine in 80% MeCN, stored at $-20\,°C$) and an appropriate amount of zirconia beads (0.1 mm) was added. The thawed cecal samples were transferred to the prepared tubes and the samples homogenized by bead beating for two minutes. The samples were stored on ice for two minutes before bead beating was repeated. Post homogenization, the samples were centrifuged twice at $13,000 \times g$, $4\,°C$ for ten minutes. From each sample, 900 µl supernatant was removed in two 450 µl portions, and each portion was added to separate 1.5 mL Eppendorf tubes. All samples were concentrated using a CentriVap fitted with a $-80\,°C$ cold trap (Labconco, Kansas, MO, USA). The dry extracts were stored at $-80\,°C$ until processed further for LC-MS/MS analysis. The dried extracts were reconstituted in 100 µl 80% MeCN containing 10 ng/ml trimethoprim (TMP) as internal standard (IS) for normalization.

### LC-MS/MS Analysis

To determine the ampicillin content, 1 µl of the reconstituted extract was analyzed by using a 1290 Infinity II LC System (Agilent Technologies, Santa Clara, CA, USA) coupled to an AB SCIEX QTrap 6500 triple quadrupole mass spectrometer (AB SCIEX Germany GmbH, Darmstadt, Germany). The samples were separated on a Gemini® 3 µm NX-C18 column (110 Å, 50 × 2 mm; Phenomenex, Torrance, CA, USA). Samples were analyzed in positive mode (electrospray ionization; ionization voltage: 4.5 kV; source temperature: $400\,°C$) using solvent A (water supplemented with 0.1% formic acid) and solvent B (acetonitrile supplemented with 0.1% formic acid). The elution was run at a flow rate of 0.7 ml/minute as follows: 5% B from 0 to 1 min followed by a linear gradient from 5% B to 95% B from 1 to 5 mins, keeping this composition (95% B) for the final minute (5 to 6 mins). Ampicillin, caffeine, and trimethoprim were detected as $[M + H]^+$ ions using a multiple reaction monitoring (MRM) experiment, and the respective transitions are listed in (Supplementary Data 2). The raw data was analyzed using the Skyline software package (version 22.2.0527) and subsequently exported to Microsoft Excel (Supplementary Data 2) for weight normalization.

### Statistical analysis

Experimental results were analyzed for statistical significance using GraphPad Prism v9.1 (GraphPad Software Inc.). Differences were analyzed by Mann-Whitney, Kruskal-Wallis, Mantel-Cox test, two-way ANOVA, Dunn's multiple comparisons, or Tukey's multiple comparisons with various post-hoc tests indicated in each figure legend. *P*-values indicated were calculated by a non-parametric Mann-Whitney *U* test or Kruskal-Wallis test comparison of totals between groups. *P*-values lower than 0.05 were considered significant: *$p < 0.05$, **$p < 0.01$, ***$p < 0.001$, ****$p < 0.0001$. A description of each statistical test can be found in the figure legends.

### Reporting summary

Further information on research design is available in the Nature Portfolio Reporting Summary linked to this article.

## Data availability

16S rRNA gene sequencing and bacterial strain genome sequencing data have been deposited in the ENA (European Nucleotide Archive, https://www.ebi.ac.uk/ena) under the accession number: PRJEB74255 and PRJEB42167, respectively. Furthermore, we have deposited our LC-MS/MS data in the MassIVE repository (https://massive.ucsd.edu) under the accession number: MassIVE ID MSV000095337. Source data are provided in this paper.

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

## Acknowledgements

We thank the staff of the animal facility and the "Genome Analytics Core Facility" of the HZI for technical support. The project was supported by the federal state Saxony-Anhalt and the European Structural and Investment Funds (ESF, 2014–2020, project number 44 100 32 030 ZS/2016/08/80645 to L.O. and T.S.), by the Joint Programming Initiative on Antimicrobial Resistance (project number 01KI1824 to T.S.), the BMBF (DF-AMR2: project number 01KI2131 to T.S.), the German Center for Infection Research (project number 06.826 to T.S.), and the Deutsche Forschungsgemeinschaft (DFG, German Research Foundation - EXC 2155 - project number 390874280 to T.S.). The funders did not influence study design, data collection and analysis, or the publishing process.

## Author contributions

E.A., L.O., and T.S. designed the study. E.A., L.E., and N.K. performed animal experiments to which M.W. provided support. E.A. and L.E. created genetically edited *K. oxytoca* and *E. coli* strains. E.A., L.E., and N.K. performed ex vivo and in vitro assays. E.A. and L.E. generated figures. C.T. provided essential feedback during experimental design, investigation, and data analysis. A.A.B. and A.G. provided technical support during sample preparations for 16 rRNA gene amplicon sequencing and metagenome sequencing. A.C.V. performed LC-MS/MS measurements and performed data analysis. T.R.L. and U.M. provided an amplicon 16S rRNA gene sequencing pipeline and conducted data analysis. T.R.L. analyzed metagenomic data, annotated KEGG and CAZyme functionalities, and contributed to data visualization. M.N.S. performed LC-MS/MS measurements and data analysis. M.B. and M.N.S. provided reagents and equipment for the LC-MS/MS measurements. E.A. and T.S. wrote and edited the manuscript incorporating input from the co-authors. M.B. and T.S. acquired funding and supervised the study.

## Funding

## Competing interests

T.S., L.O., and M.W. filed a patent for the use of *K. oxytoca* to decolonize MDR Enterobacteriaceae from the gut (EP4259171A1, EP4011384A1, WO002022122825A1 & US020240041950A1). The remaining authors declare no competing interests.
