## [Transparent Peer Review file · Nature Communications]

Klebsiella oxytoca facilitates microbiome recovery via antibiotic degradation and restores colonization resistance in a diet-dependent manner

Corresponding Author: Professor Till Strowig

Version 0:

Reviewer comments:

Reviewer #1

(Remarks to the Author)

Almási et al. investigate the interplay between the effects of host diet and antibiotic perturbation on the ability of *Klebsiella oxytoca* to outcompete and decolonize the pathogenic *Klebsiella pneumoniae*. Using in vivo experiments, the authors show that pre-colonization of mice with *K. ox* results in a significant decrease in *K. pn* abundance just 1 day after infection, in a diet-dependent manner. Mice fed standard chow and synthetic diets rich in starch or sucrose showed decolonization of *K. pn* by *K. ox*, while *K. pn* remained at high abundance in mice fed a high-sucrose/high-fat diet, despite the presence of *K. ox*. Notably, the effect of the diet was also dependent on the commensal microbial community; when performed in a germ-free mouse set-up, there was no decolonization of *K. pn*, and *K. ox* and *K. pn* were able to co-exist. The authors also analyse the effect of *K. ox* on the commensal microbiome and show that pre-colonization with *K. ox* resulted in quicker recovery to a pre-perturbation-like community and in the prevention of the outgrowth of pathobionts. Using metagenomics, the authors further show that the high-fat high-sucrose diet resulted in a significantly different KEGG functionality and was enriched for bacteria capable of degrading host-derived glycans. Finally, the authors propose a potential explanation for the observations of *K. ox* pre-colonization causing a clearance of *K. pn* in a diet- and microbiome-dependent manner. They found that *K. ox* could degrade the antibiotic, allowing the commensal microbes to recover quickly, thus enabling the decolonization of *K. pn*. While initially, direct competition between *K. ox* and *K. pn* could cause reduction in *K. pn*, in the longer term, the antibiotic degradation of *K. ox* is necessary for the microbiota to recover and enable the decolonization of *K. pn*.

The results described here are novel, with promising clinical relevance. The effects of host diet on pathogen clearance and colonization resistance have not been the focus, as most in vivo studies, standard chow is used, despite extensive knowledge on the diet-dependence of the gut microbiome. The observation of *K. pn* decolonization by *K. ox* after antibiotic perturbation being diet-dependent would also be of great interest clinically. Over all, the manuscript is well written, and the experiments well designed. I only have a few comments regarding the interpretations of results and a couple of questions.

Major comments:

The authors demonstrated that a mutant of *K. ox* that was incapable of using sucrose in vitro showed similar ability to clear *K. pn* as WT *K. ox*. They use this to claim that 'sucrose metabolism does not contribute to decolonization of *K. pn* from the gut.' I do not think the results shown justify the claim made. Other sucrose utilization mechanisms besides the PTS SacX may exist in *K. ox*, which are expressed in vivo. Furthermore, the in vitro experiment performed does not necessarily rule out the possibility that the mutant is capable of sucrose utilization in vivo. As such, these results do not rule out direct competition between *K. pn* and *K. ox* for sucrose in vivo. I would suggest the authors tone down this claim to better suit the observations, or at least speculate on alternatives in the discussion section.

While nutrient competition is a known mechanism by which probiotics decolonize pathogens and it therefore is important to focus on it, it may also be interesting to take an exploratory approach to find other potential mechanisms. The difference in KEGG functionalities between samples treated with *K. ox* and HF-HS diet, as opposed to those treated with the other diets, is interesting. What KEGG pathways other than those associated with carbohydrate metabolism were different in these samples?

For all ex vivo experiments conducted on cecal contents, including the competition assays (Figure 3b) and the *L. reuteri*

growth assays (Figure 6c), it is unclear how many independent mice were involved in these experiments. Specifically, it's unclear whether the cecum of one individual mouse was considered as one biological replicate. Or were the cecal contents pooled? This clarification is vital for evaluating both the reproducibility and variability of measurements, which in turn directly impact the reliability of the presented results and consequently, the conclusions drawn by the authors.

Minor comments:

Figure 4b: Tiles represent alpha diversity, right? Please indicate in figure legends, similar to AUC in Figure 6e.

Figure 4: Please standardize the labelling of the experimental groups: + A + K.o. (in panel B), is the same as Chow + K. oxy (in panel d) and + amp + K. oxy (in panels e, f, g), correct?

Line 266: "... a plasmid carrying a beta-lactamase gene (pGEX) to induce ampicillin tolerance". It is unclear to me why this is named tolerance and not resistance?

Figure 6D and 6F: Are these representative growth curves? How many replicates were performed? What is shown? Mean, median?

Figure 6e: AUCs are based on how many replicates? What baseline correction was used? This part is not covered at all in the Methods section.

Extended data Table 1:

- Tab LEfSe_MAGs: Columns B, D, E need titles, in J1: comment "needs input from Till Robin": please clarify and remove.
- Tab LEfSe_CAzymes: Add titles and remove question marks

Reviewer #2

(Remarks to the Author)

In this manuscript the authors describe how *Klebsiella oxytoca* impacts *Klebsiella pneumoniae* intestinal colonisation in mice in the context of four different diets: standard chow, high starch, high sucrose, or high fat/high sucrose. The authors showed that *K. oxytoca* pre-colonisation degraded ampicillin in the intestine, which allowed for recovery of the gut microbiota in a diet-dependent manner that suppressed *K. pneumoniae* growth. This is a very interesting study, however I do have several comments and some questions regarding the experimental design:

- 1) Ampicillin is not really used clinically anymore due to the rise in pathogen resistance to this antibiotic. Many studies have described other broad-spectrum antibiotics that are known to promote intestinal colonisation of carbapenem-resistant *Klebsiella* (e.g. carbapenems). Why have the authors chosen to test ampicillin?
- 2) *K. oxytoca* is pathogenic, and I am very doubtful that this strain would ever be approved for clinical use to treat *K. pneumoniae* intestinal colonisation. This paper even describes that this strain of *K. oxytoca* has beta-lactamase activity. Moreover, *K. oxytoca* is not often found in healthy adult gut microbiota, indicating that they are not essential for colonisation resistance in healthy people. Why select this strain to test in this paper over other safer gut commensals?
- 3) Why is a high fat/high sucrose diet and a high sucrose diet tested, but not a high fat diet?
- 4) In the ex vivo competition assays described in lines 150-156, *K. oxytoca* and *K. pneumoniae* are not inoculated equally – why? What happens if they are inoculated equally? From what I understand, in the germ-free mouse competition experiments described in lines 156-163 the *K. oxytoca* and *K. pneumoniae* were inoculated equally and showed different results from the ex vivo competition assays?
- 5) The 16S sequencing data described in lines 167-204 is difficult to properly interpret without incorporating changes in biomass. Antibiotics can reduce bacterial biomass by killing susceptible commensals, but resistant strains can bloom during antibiotic treatment. It is unclear how these diets could potentially impact the total bacterial biomass as well. Therefore, in a relative abundance plot as it is currently presented, a particular taxa that is present at a high relative abundance could either have bloomed (increased in absolute abundance) or stayed the same (if other bacteria were killed and there was a decrease in total biomass).
- 6) The beta-lactamase activity of *K. oxytoca* appears to be important for promoting recovery of the gut microbiota to inhibit *K. pneumoniae* growth. This would be better demonstrated by knocking out the beta-lactamase activity of *K. oxytoca* then adding beta-lactamase to *E. coli*, as this will shift other parameters (e.g. there may be a difference in nutrient utilisation between these strains).
- 7) Overall this paper has demonstrated that *K. oxytoca* degrades antibiotics that promotes recovery of the gut microbiota, but not how the gut microbiota decolonises *K. pneumoniae* from a mechanistic perspective?
- 8) In the figure legends it is not always clear how many mice are tested per group, and how many independent experiments are performed.
- 9) It appears that the sex of the mice was matched in each treatment group, however it is not clear overall how many male and female mice were tested in each treatment group.
- 10) Lines 311-326 of the discussion spend too much time re-summarising results.
- 11) Lines 330-332 should be expanded upon in the discussion.
- 12) Lines 349-351 should also be expanded upon in the discussion.
- 13) Line 371-373: What was the minimum number of mice per cage?
- 14) Line 374: Should say sex not gender.
- 15) Lines 380-384: The exact composition of the diets is not clear. What is in the standard chow diet? What else is in the high sucrose diet – presumably this is not 100% sucrose? Also, these diets described here do not match the descriptions of

the diets tested in this paper (there was no fat only diet tested, and there is no description of the high starch diet)?

16) Lines 387-389: Are these patient isolates well characterised? Have the strains been deposited? Have their genomes been sequenced and has this sequencing data been deposited?

17) Lines 398-400: Why were mice gavaged both orally and rectally, instead of just orally as is standard practice?

18) Line 435: Should this say M9 media not MM9 media? And was this performed aerobically with shaking?

19) Shotgun metagenome sequencing data and LC-MS/MS data should be deposited into a public repository.

20) Line 535: Why was caffeine and trimethoprim measured?

Reviewer #3

(Remarks to the Author)

Reviewer #4

(Remarks to the Author)

The authors aimed to recapitulate commensal-pathogen dynamics in a murine model of co-colonisation that is representative of westernised human diets of high starch, high sucrose, and high fat/high sucrose. The authors focus on the interaction between *Klebsiella oxytoca* and *Klebsiella pneumoniae* and employ a combination of amplicon 16S rRNA gene sequencing, metagenomic sequencing, mass spectrometry, and a suite of in vitro and in vivo experiments. I commend the authors on bringing together a well rounded study.

I have both minor and major comments.

My main concern is with regards to the analysis of amplicon 16S rRNA gene data; namely, using QIIME1 for analysis here instead of QIIME2. QIIME1 is no longer supported and has been replaced by QIIME2 since the beginning of 2018. There are key changes between the two approaches in how microbial diversity is now calculated – i.e., calling amplicon sequence variants (ASVs) using QIIME2 which resolves single nucleotide variants, compared to clustering operational taxonomic units (OTUs) in QIIME1 at 97% similarity. This has implications in determining and quantifying compositional changes, and identifying key microbial taxa involved. Additionally, it was not clear if samples were rarefied prior to alpha and beta diversity calculations?

I would recommend recomputing microbiota analyses using QIIME2 and confirming similar conclusions can be made from this study before I would be comfortable recommending this manuscript for acceptance into Nature Communications. I think the quality of sequencing (i.e., using PE300) lends well to merging R1 and R2 reads for the V4 hypervariable region as the overlap here would be greater than 50bp, but it is not necessary to merge (c.f., <https://academic.oup.com/nar/article/35/18/e120/2401970>). When using QIIME2, I would suggest using *deblur* instead of *dada2* (especially if keeping with merged reads) to determine ASVs.

Some very minor corrections by editorial team throughout the text of the manuscript e.g.,

Line 61: I think "both" is misused as only 'systemic infection' is provided.

Line 114: revise dependent to independent

Line 196: sentence needs a rewrite: "However, HS diet showed a unique expansion of at Staphylococcaceae a group average of 37% in early post-ampicillin recovery."

Line 309: revise to: "Amplicon 16S rRNA gene sequencing revealed.."

Line 409: revise to: "LB agar plates supplemented with 50 ug/ml kanamycin for *K. pneumoniae* selection and quantification, and MacConkey agar plates for *K. oxytoca* CFU quantification."

Please clarify if referring to microbial composition or diet content in line 313: "Since the composition of the three semi-synthetic diets differs fundamentally in sucrose and fat content, the shared taxonomic changes are most probably attributed to the lack of complex dietary fibers in all three diets."

As composition(al) is already being used to describe the microbial community, I would suggest revising line 313 to the effect of: "since the contents of the three semi-synthetic diets.."

Line 333: "Ex vivo assessment of intestinal content collected from ampicillin-treated mice with or without *K. oxytoca* pre-colonisation..."

Line 404: please provide the make and model of the mechanical lysis machine, and the value and units for the 'speed' setting.

Line 452: speed of mechanical lysis? And for all instances of mechanical lysis

Line 454 and throughout: "Next, homogenized samples were centrifuged at 8,000 xg for 3 minutes."

Line 431: what were the volume of cultures used?

Line 454 and throughout: Consistency with using period or comma to denote numbers in the thousands.

Line 469: please provide primer sequences.

Line 469: please revise to read: "Amplification of the V4 hypervariable region of the 16S rRNA gene was performed..."

Line 483: replace "OTU absolute abundance" with "OTU frequency table".

It's great to see the amplicon 16S rRNA gene data being made publicly available. It would also be great to see the shotgun metagenomic and metabolomics data being made available too.

Extended data:

LefSe for MAGs and CAZymes have "???" in column headers – needs to be addressed

Some references in text still read PMID

Version 1:

Reviewer comments:

Reviewer #1

(Remarks to the Author)

The authors have done an very good job addressing my feedback. The manuscript is now well-suited for publication in Nature Communications.

Reviewer #2

(Remarks to the Author)

I appreciate the steps the authors have taken to clarify points of their paper and to perform additional experiments to address some comments. However, unfortunately there are some fundamental aspects of this paper that I disagree with that I do not think can be addressed in a revised version of the manuscript:

(1) The authors concluded that *K. oxytoca* protects the host by degrading ampicillin, allowing for recovery/protection of the gut microbiome, and it is gut microbiome recovery that protects against *K. pneumoniae* colonization. It is already well known that a healthy gut microbiome protects against *K. pneumoniae* intestinal colonization. Moreover, it is not a novel idea to degrade antibiotics to protect against pathogen intestinal colonization. Several previously published papers have explored direct administration of an enzyme to degrade several antibiotics (including more clinically relevant antibiotics than ampicillin) demonstrating that this can be accomplished in a safer manner than administering *K. oxytoca* (e.g. reviewed here: <https://doi.org/10.1093/infdis/jiab143>). Finally, as mentioned in my previous comments, ampicillin is not relevant clinically to the group of patients that require new interventions to prevent or treat *K. pneumoniae* intestinal colonisation - immunocompromised adults. Therefore, I do not see the clinical relevance of *K. oxytoca* mediated protection described in this paper.

(2) The authors demonstrated that *K. oxytoca* is protective against *K. pneumoniae*, but as I mentioned in my previous comments an antibiotic-resistant opportunistic pathogen such as *K. oxytoca* will not be approved for clinical use in patient groups that are the most susceptible to the intestinal colonisation with *K. pneumoniae* – immunocompromised patients. I did not find the argument about *E. coli* to be convincing, as *E. coli* is often found in healthy adults unlike *K. oxytoca*. Moreover, the argument that *K. oxytoca* can asymptotically colonize infants and children is not reassuring, as another pathogen *Clostridioides difficile* can also asymptotically colonize infants but can cause disease in adults. So again, I don't see the clinical relevance of using *K. oxytoca* to prevent *K. pneumoniae* intestinal colonization, especially if safer alternatives such as engineering a commensal to degrade the antibiotic or using purified enzymes directly (which is already being explored as mentioned above) are available.

Reviewer #3

(Remarks to the Author)

Reviewer #4

(Remarks to the Author)

major:

I appreciate the authors efforts in addressing the QIIME1 vs., QIIME2 workflow. I believe the initial concerns have been addressed.

minor:

reviewer comment version 1, Line 469: please revise to read: "Amplification of the V4 hypervariable region of the 16S rRNA gene was performed.."

Author's response: The text was rephrased to "Amplification of the 16S rDNA V4 hypervariable region was performed.."

16S rDNA V4 is incorrect terminology. Should read "16S rRNA gene", which is at the DNA level.
i.e., "Amplification of the 16S rRNA gene V4 hypervariable region was performed.."

REVIEWER COMMENTS

Reviewer #1 (Remarks to the Author):

Almási et al. investigate the interplay between the effects of host diet and antibiotic perturbation on the ability of Klebsiella oxytoca to outcompete and decolonize the pathogenic Klebsiella pneumoniae. Using in vivo experiments, the authors show that pre-colonization of mice with K. ox results in a significant decrease in K. pn abundance just 1 day after infection, in a diet-dependent manner. Mice fed standard chow and synthetic diets rich in starch or sucrose showed decolonization of K. pn by K. ox, while K. pn remained at high abundance in mice fed a high-sucrose/high-fat diet, despite the presence of K. ox. Notably, the effect of the diet was also dependent on the commensal microbial community; when performed in a germ-free mouse set-up, there was no decolonization of K. pn, and K. ox and K. pn were able to co-exist. The authors also analyse the effect of K. ox on the commensal microbiome and show that pre-colonization with K. ox resulted in quicker recovery to a pre-perturbation-like community and in the prevention of the outgrowth of pathobionts. Using metagenomics, the authors further show that the high-fat high-sucrose diet resulted in a significantly different KEGG functionality and was enriched for bacteria capable of degrading host-derived glycans. Finally, the authors propose a potential explanation for the observations of K. ox pre-colonization causing a clearance of K. pn in a diet- and microbiome-dependent manner. They found that K. ox could degrade the antibiotic, allowing the commensal microbes to recover quickly, thus enabling the decolonization of K. pn. While initially, direct competition between K. ox and K. pn could cause reduction in K. pn, in the longer term, the antibiotic degradation of K. ox is necessary for the microbiota to recover and enable the decolonization of K. pn. The results described here are novel, with promising clinical relevance. The effects of host diet on pathogen clearance and colonization resistance have not been the focus, as most in vivo studies, standard chow is used, despite extensive knowledge on the diet-dependence of the gut microbiome. The observation of K. pn decolonization by K. ox after antibiotic perturbation being diet-dependent would also be of great interest clinically. Over all, the manuscript is well written, and the experiments well designed. I only have a few comments regarding the interpretations of results and a couple of questions.

Major comments:

1. The authors demonstrated that a mutant of K. ox that was incapable of using sucrose in vitro showed similar ability to clear K. pn as WT K. ox. They use this to claim that 'sucrose metabolism does not contribute to decolonization of K. pn from the gut.' I do not think the results shown justify the claim made. Other sucrose utilization mechanisms besides the PTS SacX may exist in K. ox, which are expressed in vivo. Furthermore, the in vitro experiment performed does not necessarily rule out the possibility that the mutant is capable of sucrose utilization in vivo. As such, these results do not rule out direct competition between K. pn and K. ox for sucrose in vivo. I would suggest the authors tone down this claim to better suit the observations, or at least speculate on alternatives in the discussion section.

Author's response: After careful consideration of the reviewer's comment, we changed the title of the Result section 'Sucrose utilization of K. oxytoca does not contribute to decolonization of K. pneumoniae in vivo' to 'Sucrose utilization gene sacX of K. oxytoca does not contribute to decolonization of K. pneumoniae in vivo' to interpret our results more narrowly, only related to

the effect of the gene *sacX* in the in vivo setting rather than sucrose utilization generally in interspecies competition. We made several further modifications to the main text in this section, reflecting the above-mentioned revised interpretation (see lines 126-150).

2. While nutrient competition is a known mechanism by which probiotics decolonize pathogens and it therefore is important to focus on it, it may also be interesting to take an exploratory approach to find other potential mechanisms. The difference in KEGG functionalities between samples treated with *K. ox* and HF-HS diet, as opposed to those treated with the other diets, is interesting. What KEGG pathways other than those associated with carbohydrate metabolism were different in these samples?

Author's response: Following the reviewer's suggestion, we followed an exploratory approach to identify KEGG pathways enriched in the microbial communities in the mice that cleared *K. pneumoniae*, i.e. 'clearer' microbial communities, compared to 'non-clearer' ones, which failed to clear *K. pneumoniae*. As introduced in our manuscript, the 'clearer' group comprised *K. oxytoca* pre-colonized mice on chow, HSt, and HS diets. At the same time 'non-clearers' included all ampicillin-treated mice without *K. oxytoca* colonization and HF/HS *K. oxytoca* pre-colonized mice. The revised Extended Data Figure 6 and Extended Data Table 3 include the new analyses. In 'clearer' samples, we identified various enriched KEGG modules and pathways related to the utilization of carbohydrates (including glycosaminoglycans and lipopolysaccharides) and amino acids. Notably, we found enrichment of multiple functional groups related to drug resistance and degradation of aromatic compounds in the 'Clearer' group, which could be related to ampicillin treatment. 'Non-clearer' groups also showed enrichment of a single functional module linked to drug resistance, as well as multiple functional groups representing central carbohydrate metabolism, which might be explained by the blooming of bacteria with pathogenic potential such as Enterococcaceae and Staphylococcaceae favoring simple carbon sources as opposed to complex fibers. Additionally, we found an enrichment of different functional groups representing cofactor and vitamin metabolism in both 'clearers' and 'non-clearers'. The results are briefly described in lines 231-241. This more comprehensive bioinformatics analysis complements our past analysis of CAZymes in particular, yet, we have not conducted any additional wet-lab analysis, which we consider to be outside our manuscript's scope.

3. For all ex vivo experiments conducted on cecal contents, including the competition assays (Figure 3b) and the *L. reuteri* growth assays (Figure 6c), it is unclear how many independent mice were involved in these experiments. Specifically, it's unclear whether the cecum of one individual mouse was considered as one biological replicate. Or were the cecal contents pooled? This clarification is vital for evaluating both the reproducibility and variability of measurements, which in turn directly impact the reliability of the presented results and consequently, the conclusions drawn by the authors.

Author's response: We carefully revised the manuscript to avoid misunderstandings about the technical details of the ex vivo experiments. Competition experiments between *K. oxytoca* and *K. pneumoniae*, shown in Figure 3b, were performed in caecum content pooled from multiple mice (n=3-6 mice). This pooled cecal content was then used in independent experiments, i.e., with different cultures of bacteria to assess competition. We explicitly rephrased the Method section "Ex vivo competition assay between *K. pneumoniae* and *K. oxytoca*" to include this detail. Moreover, the legend of Figure 3a now specifies that caecum contents were pooled, and we changed the phrase "biological replicate" to "independent experiment" to avoid

confusion with experiments shown in Figure 6. In the “outgrowth” experiments with *L. reuteri* I49 and *E. coli* MG1655, we investigated the inhibitory potential of caecum contents from each mouse separately; these data points we consider biological replicates (= each of them is content from a different mouse). Accordingly, we changed the legend of Figures 6c, d, e, and f to state explicitly that caecum contents from individual mice were kept separate. This we further highlighted in the respective method sections “*Ex vivo* assessment of β -lactamase activity” and “*Ex vivo* outgrowth assay with *L. reuteri* I49 and *E. coli* MG1655 and subsequent data analysis”.

Minor comments:

- Figure 4b: Tiles represent alpha diversity, right? Please indicate in figure legends, similar to AUC in Figure 6e.

Author’s response: We specified the legend of Figure 4b accordingly.

- Figure 4: Please standardize the labelling of the experimental groups: + A + K.o. (in panel B), is the same as Chow + K. oxy (in panel d) and + amp + K. oxy (in panels e, f, g), correct?

Author’s response: We standardized the labeling across different panels of Figure 4: “C” marks the control group, “Amp” marks ampicillin-treated mice, and “Amp+K.o.” marks ampicillin-treated and *K. oxytoca*-colonized mice.

- Line 266: “... a plasmid carrying a beta-lactamase gene (pGEX) to induce ampicillin tolerance”. It is unclear to me why this is named tolerance and not resistance?

Author’s response: “Tolerance” was replaced with “resistance” in the marked sentence.

- Figure 6D and 6F: Are these representative growth curves? How many replicates were performed? What is shown? Mean, median?

Author’s response: Additional technical details were added to the legends of Figures 6d and 6f. Furthermore, we included the standard deviations of the depicted biological replicates:

Figure 6d: “(d) *L. reuteri* I49 growth in a mixture of (ratio 1:1) caecum content supernatants and MRS (final concentration: 1x) for 48 hours. Displayed growth curves represent three to four biological replicates with standard deviation. Each biological replicate represents caecum content supernatant obtained from an individual mouse. Growth curves were recorded in three technical replicates, which were averaged before data visualization. P-values on the right side represent Dunn’s multiple comparisons of the biological replicates in each condition at 48 hours with * $p < 0.05$, ** $p < 0.01$, *** $p < 0.005$, **** $p < 0.0001$.”

Figure 6f: “(f) *E. coli* MG1655 growth in a mixture of caecum content supernatants and LB (final concentration 1:1) for 48 hours. Displayed growth curves represent three biological replicates with standard deviation. Each biological replicate represents caecum content supernatant obtained from an individual mouse. Growth curves were recorded in three technical replicates, which were averaged before data visualization.”

- Figure 6e: AUCs are based on how many replicates? What baseline correction was used? This part is not covered at all in the Methods section.

Author's response: Additional technical details were added to the legends of Figure 6e. Furthermore, we provide more technical details in the method section "Ex vivo outgrowth assay with *L. reuteri* I49 and *E. coli* MG1655 and subsequent data analysis" (lines 534-545).

Figure 6e: "(e) Heatmap depicting *L. reuteri* I49 growth represented by area under the curve (AUC) measurements after 48 hours in a mixture of (ratio 1:1) caecum content supernatants and MRS (final concentration: 1x). AUC values were calculated for each biological replicate based on the mean OD₆₀₀ values of three technical replicates, blanked to the average of the first two measurements. Thus, baseline correction was not used for AUC computing."

Ex vivo outgrowth assay with *L. reuteri* I49 and *E. coli* MG1655 and subsequent data analysis: "Caecum contents were isolated from individual mice as described in the previous section. Next, sterilized and serially diluted caecum content supernatants (with PBS from 1:2 to 1:256) were supplemented 1:1 with 2X MRS and 2X LB liquid media to support the growth of *L. reuteri* I49 and *E. coli* MG1655, respectively. OD₆₀₀=0.8 *L. reuteri* I49 or *E. coli* MG1655 culture was inoculated in 1:50 ratio and incubated under aerobic conditions for 48 hours with continuous shaking. OD₆₀₀ values were recorded hourly for 3-4 biological replicates (representing caecum content from individual mice) and three technical replicates. The mean of the three technical replicates was normalized to the mean of the first two measurements used as baseline. AUC calculations for *L. reuteri* I49 growth were performed using GraphPad Prism based on the 3-4 biological replicates. Since the data was already normalized in previous steps, a baseline of Y=0 was applied."

- Extended data Table 1:
 - Tab LEfSe_MAGs: Columns B, D, E need titles, in J1: comment "needs input from Till Robin": please clarify and remove.
 - Tab LEfSe_CAZymes: Add titles and remove question marks

Author's response: The notes were removed from Extended Data Table 1, and titles were added accordingly.

Reviewer #2 (Remarks to the Author):

In this manuscript the authors describe how *Klebsiella oxytoca* impacts *Klebsiella pneumoniae* intestinal colonisation in mice in the context of four different diets: standard chow, high starch, high sucrose, or high fat/high sucrose. The authors showed that *K. oxytoca* pre-colonisation degraded ampicillin in the intestine, which allowed for recovery of the gut microbiota in a diet-dependent manner that suppressed *K. pneumoniae* growth. This is a very interesting study, however I do have several comments and some questions regarding the experimental design:

1. Ampicillin is not really used clinically anymore due to the rise in pathogen resistance to this antibiotic. Many studies have described other broad-spectrum antibiotics that are known to promote intestinal colonisation of carbapenem-resistant *Klebsiella* (e.g. carbapenems). Why have the authors chosen to test ampicillin?

Author's response: The reviewer is correct that various broad-spectrum antibiotics have been suggested to influence human colonization with carbapenem-resistant *Klebsiella pneumoniae* strains. For research purposes, simplified animal models of reduced colonization resistance are widely used, either intrinsically in the case of gnotobiotic mouse models or after antibiotic disruption of the microbiota. While the *K. pneumoniae* strain we used in this study is multidrug-resistant, the *K. oxytoca* isolate tested as a probiotic candidate is not. It only encodes an

endogenous beta-lactamase, hence our choice of the ampicillin-mediated disruption of the microbiota.

Nevertheless, especially in the pediatric population, ampicillin and the closely related antibiotic amoxicillin are still considered relevant, often a suggested first-line treatment option. As a result, we believe that our mouse model is relevant to human health and has enabled us to test the competitiveness of a wide range of commensal isolates (see also Osbelt et al., *Cell Host Microbe* 2021 (PMID: 34610293), Osbelt et al., *Nat Microbiol* 2024 (PMID: 38862602)).

2. *K. oxytoca* is pathogenic, and I am very doubtful that this strain would ever be approved for clinical use to treat *K. pneumoniae* intestinal colonisation. This paper even describes that this strain of *K. oxytoca* has beta-lactamase activity. Moreover, *K. oxytoca* is not often found in healthy adult gut microbiota, indicating that they are not essential for colonisation resistance in healthy people. Why select this strain to test in this paper over other safer gut commensals?

Author's response: We agree that the safety of each probiotic candidate should be investigated as carefully as its efficacy. For species in the Enterobacteriaceae family, it is particularly difficult to separate purely "pathogenic" and purely "beneficial" strains from one another. A commonly cited example is *E. coli*, a species representing enormous strain heterogeneity, several of which are causative agents of severe human infections, and others are clinically relevant probiotics. Specifically, *E. coli* Nissle has been used as a probiotic for over a century, with only recent reports focusing on possible adverse effects to humans, i.e., through the production of colibactin (Nougayrède et al., *mSphere* 2021 (PMID: 34378987), Pleguezuelos-Manzano et al., *Nature* 2020 (PMID: 32106218)). Similarly, specific *K. oxytoca* isolates and also other strains belonging to the *Klebsiella oxytoca* species complex have been linked to antibiotic-induced hemorrhagic colitis (Schneditz et al., *PNAS* 2014 (PMID: 25157164)). However, the link of gut colonization to disease induction appears relatively low as many healthy infants and children are asymptotically colonized with members of the *Klebsiella oxytoca* species complex (Greimel et al., *JPGN* 2021 (PMID: 34520403)). While the *K. oxytoca* MK01 strain was isolated from a healthy infant and it caused no harm in mice lacking adaptive immune cells (Osbelt et al., *Cell Host Microbe* 2021 (PMID: 34610293)), further studies into its safety, comparable to what has been described for the related *Klebsiella* ARO112 strain in a recent preprint (Cabral et al., *bioRxiv* 2023), are planned.

Additionally, concerning the reviewer's point about the low prevalence of *K. oxytoca* strains in the healthy adult population, we agree with the reviewer's assessment of the prevalence but think that the low prevalence could also speak to the benefit of using *K. oxytoca* as a potential probiotic. Importantly, when discussing the use of next-generation probiotics, biocontainment is repeatedly discussed (Mandell et al., *Nature* 2015 (PMID: 25607366)). In an ideal scenario when *K. oxytoca* is utilized as a specific competitor against *K. pneumoniae*, the probiotic strain would be eliminated from the host after pathogen clearance rather than engraft permanently. Similarly to our *in vivo* observations, the levels of *K. oxytoca* reduce steadily following the cessation of antibiotic treatment and, in some animals, start clearing out from the colon soon after *K. pneumoniae* clearance. We consider this to be a positive feature that potentially calls for little to no additional efforts to achieve biocontainment.

3. Why is a high fat/high sucrose diet and a high sucrose diet tested, but not a high fat diet?

Author's response: In the current study, we aim to investigate microbial competition for carbohydrates in different dietary contexts relevant to human health. The high-fat/high-sucrose diet utilized in this study was designed to mimic a Western-style diet that induces

metabolic syndrome and obesity. The high-starch and high-sucrose diets served as control diets to the high-fat/high-sucrose diet and helped dissect the effect of (1) replacing a complex mixture of plant-derived fibers with starch, (2) the addition of sucrose, and (3) the addition of animal-derived fats. Furthermore, since about 2/3 of the dietary caloric content of an optimal rodent diet is carbohydrate-derived, it is incredibly difficult to reduce the carbohydrate content of a feed beyond a certain threshold. Even though the caloric fat content of a feed is often at around or below 10%, even an extremely high-fat feed still contains around 20 kJ% of carbohydrates. Hypothetically, the utilization of a high-fat/low-sucrose diet would have been considered, although a substantial ratio of the caloric content of that diet would have derived from starch. In the current study, the three semi-synthetic diets, in comparison with the standard chow diet, helped us investigate carbohydrate competition in different dietary regimens, to which the inclusion of a “high-fat/high-starch” diet would not have provided additional insights in our opinion.

4. In the *ex vivo* competition assays described in lines 150-156, *K. oxytoca* and *K. pneumoniae* are not inoculated equally – why? What happens if they are inoculated equally? From what I understand, in the germ-free mouse competition experiments described in lines 156-163 the *K. oxytoca* and *K. pneumoniae* were inoculated equally and showed different results from the *ex vivo* competition assays?

Author’s response: An important difference between our *ex vivo* and *in vivo* competition assays is the timing of bacterial inoculation. While mice are always pre-colonized with *K. oxytoca* before *K. pneumoniae* inoculation, an *ex vivo* setting is better suited for simultaneous inoculating the two strains. To better model the *in vivo* competition assay, we follow a 10:1 *K. oxytoca*:*K. pneumoniae* inoculation regimen to mimic the advantage of an already stable *K. oxytoca* colonization in the mouse gut. Furthermore, in our experience, the 10:1 commensal:pathogen ratio has a higher potential to predict *in vivo* competitiveness (see also Osbelt et al., Nat Microbiol, 2024 (PMID: 38862602)). Non-protective commensal strains do not result in significant pathogen reduction even at 10:1 ratios, while protective ones facilitate a more pronounced CFU reduction of the pathogenic strain compared to equal inoculation. To reiterate, we use an *ex vivo* germ-free competition assay as a preliminary screening for potential pathogen reduction, which is then confirmed *in vivo*. Of note, the caecum content of germ-free mice is highly enriched in easily accessible nutrients and carbon sources, and it is rarely growth-limiting to two bacteria.

Nevertheless, to demonstrate the impact of inoculation ratios on the CFU reduction, we repeated our *ex vivo* assay with four different commensal to pathogen ratios, namely 1:1, 2:1, 5:1, and 10:1 *K. oxytoca*:*K. pneumoniae*. The results are summarized in the Reviewer Figure 1 below. Only dilutions $\geq 5:1$ and 10:1 for the HF/HS diet resulted in statistically significant *K. pneumoniae* CFU reduction using the Kruskal-Wallis non-parametric test with Dunn’s multiple comparisons between the four ratios and the control (*K. pneumoniae* only). The outcome of the statistical analysis echoes the observations throughout the manuscript, namely, that *K. pneumoniae* CFU reduction was most potent on the Chow diet compared to all semi-synthetic diets. This data is provided to the reviewers, but at this point, we would not include it in the manuscript.

Reviewer Figure 1: Ex vivo competition assay between *K. oxytoca* and *K. pneumoniae* in GF caecum content: CFUs of *K. pneumoniae* after 24 hours of co-culturing with *K. oxytoca* in GF caecum content from mice on (a) Chow, (b) high-starch, (c) high-sucrose or (d) high-fat/high-sucrose diets. Bars represent averages of 4 independent experiments using different cultures of bacteria. Dots represent average of three technical replicates. P-values indicated at individual time points represent Mann-Whitney non-parametric rank comparison with * $p < 0.05$, ** $p < 0.01$, *** $p < 0.005$, **** $p < 0.0001$. Upper dotted line represents the average of control group CFUs, lower dotted lines depict a 10-fold reduction of CFUs from control. Dots represent biological replicates, each of which are means of 3 technical replicates. Caecum content was isolated from GF mice fed one of four diets for 14 days, pooled, and used as a medium base for competition assays between *K. oxytoca* and *K. pneumoniae*.

- The 16S sequencing data described in lines 167-204 is difficult to properly interpret without incorporating changes in biomass. Antibiotics can reduce bacterial biomass by killing susceptible commensals, but resistant strains can bloom during antibiotic treatment. It is unclear how these diets could potentially impact the total bacterial biomass as well. Therefore, in a relative abundance plot as it is currently presented, a particular taxa that is present at a high relative abundance could either have bloomed (increased in absolute abundance) or stayed the same (if other bacteria were killed and there was a decrease in total biomass).

Author's response: We agree that performing 16S rRNA amplicon sequencing alone cannot determine the mass of bacterial populations in the gut. Ideally, the absolute abundance of specific bacterial strains can be quantified using strain-specific qPCR probes, or bacterial loads can be quantified via selective plating-dependent approaches. Unfortunately, we did not perform these analysis directly nor preserved corresponding samples but proceeded to DNA extraction. In response to the reviewer's comment we attempted to quantify bacterial loads using the available DNA left from these experiment. However, we lacked concentrations of

DNA per mg stool and therefore failed to obtain conclusive and reliable data regarding 16S rRNA copy numbers. As an alternative, we longitudinally quantified *K. oxytoca* loads on the four diets, so we can use Enterobacteriaceae relative abundance as a proxy for *K. oxytoca* relative abundance. Based on these data, we calculated overall bacterial loads, at least in *K. oxytoca* colonized mice (see Reviewer Figure 2 below). The results show significant variability within groups rather than diet or sampling time. Since day 12 represented a sample merely 48 hours after the cessation of ampicillin treatment, it is doubtful that overall bacterial loads throughout the experiment would have that level of consistency.

Reviewer Figure 2: *K. oxytoca* CFUs and all fecal CFUs in ampicillin-treated *K. oxytoca* colonized mice: (a) *K. oxytoca* CFUs throughout the animal experiment quantified by plating on selective agar plates. (b) All bacterial CFUs in fecal samples quantified by *K. oxytoca* CFUs and relative abundances extracted from 16S gene amplicon sequencing data.

We recognize that the absolute quantification of bacterial taxa that we describe as “blooming” would be of great interest. Nevertheless, comparing ampicillin-treated communities on different diets is still valuable and informative regarding the most abundant taxa on each dietary background. The lack of absolute quantification of bacterial loads, Enterococcaceae and Staphylococcaceae CFUs thus remains a limitation to this study. Hence, we have included a statement reflecting this limitation in the discussion, see line 333.

- The beta-lactamase activity of *K. oxytoca* appears to be important for promoting recovery of the gut microbiota to inhibit *K. pneumoniae* growth. This would be better demonstrated by knocking out the beta-lactamase activity of *K. oxytoca* then adding beta-lactamase to *E. coli*, as this will shift other parameters (e.g. there may be a difference in nutrient utilisation between these strains).

Author’s response: We agree with the Reviewer’s assessment that the benefits of direct bacterial competition and ampicillin degradation should be ideally disentangled to assess each of their contributions to *K. pneumoniae* clearance. Because our *in vivo* colonization model necessitates ampicillin resistance in both the protective and pathogenic strains, we could not create a beta-lactamase-deficient genetic mutant of *K. oxytoca* for *in vivo* colonization of SPF mice, as the Reviewer suggested.

[Redacted]

Mice were colonized with each of the strains separately following ampicillin treatment (Reviewer Figure 4a) and colonization levels were confirmed three days later, and similar fecal CFU levels were recovered (Reviewer Figure 4b). In a similar assay to main Figure 6 (d), *L. reuteri* grew equally well in cecum contents *ex vivo* from mice colonized with each of the 3 strains (Reviewer Figure 4c).

Moreover, we used LC-MS/MS to quantify residual ampicillin levels in the caecum content of mice. As observed in our previous assays, ampicillin levels quantified by weight-normalized peak areas decreased significantly (unpaired t-test with **** $p < 0.0001$), approximately 1000-fold in the caecum of pre-colonized mice compared to controls with all three strains (Reviewer Figure 4d).

This experiment demonstrates that the beta-lactamase activity of other Enterobacteria promotes ampicillin degradation to a similar extent *in vivo*, but without direct competition with the pathogenic strain, it is insufficient to promote pathogen CFU reduction and subsequent clearance.

[Redacted]

Reviewer Figure 4: *Limosilactobacillus reuteri* outgrowth assay from caecum content of mice: (a) Schematic showing in vivo experimental setup. (b) Fecal colonization of *K. oxytoca*, *K. michiganensis* and *E. coli* was confirmed by CFU quantification on day 8. (c) *L. reuteri* 149 growth in a mixture of (ratio 1:1) caecum content supernatants and MRS media (final concentration: 1x) for 48 hours. Displayed growth curves represent three to four biological replicates with standard deviation. Each biological replicate represents caecum content obtained from an individual mouse. Growth curves are represented as average from three technical replicates. (d) Weight-normalized AMP peak areas determined in cecal extracts of mice (n=4-6 mice/group) treated with ampicillin and colonized with different bacterial strains. Data are displayed as individual biological replicates (mean of double injection), bars indicate the median with 95% confidence intervals. The indicated p-values represent unpaired t-test with *p < 0.05, **p < 0.01, ***p < 0.005, ****p < 0.0001. (in c) Isolated caecum contents (n=4-6 mice/group) were diluted with 2X volume of PBS, after which they were centrifuged in three consecutive steps, removing the supernatant each time. Final supernatants were filtered using a 0.22 µm filter. 2X MRS was supplemented to support the growth of *L. reuteri* 149.

7. Overall this paper has demonstrated that *K. oxytoca* degrades antibiotics that promotes recovery of the gut microbiota, but not how the gut microbiota decolonizes *K. pneumoniae* from a mechanistic perspective?

Author's response: The reviewer is correct; in this manuscript, we have summarized our findings on how the dietary background of the host affects *K. pneumoniae* decolonization and clearance in the presence of *K. oxytoca*. In a previous study we extensively characterized the mechanistic background of the interspecies competition between these two and other strains (Osbelt et al. 2021, Cell Host & Microbe). We concluded from that study that competition for available carbon sources is of fundamental importance for *Klebsiella* species. Moreover, a wide range of accessible carbon sources is a prerequisite to effective competitiveness and pathogen clearance. Our experimental setup presented in the manuscript derive from these data. In the current manuscript, we aim to study these previously characterized competitive interactions within the context of dietary regimens representative of human Westernized diets to assess how these competitions are affected in different dietary backgrounds. Of note, in a recently published study by the group of K. Honda (Furuichi et al., Nature 2024), the authors identified indirectly a critical role of the commensal microbiota for gluconate utilization to outcompete *K. pneumoniae* from the gut. Whether gluconate is also central in the context of our model will require additional experiments beyond the scope of the current manuscript.

8. In the figure legends it is not always clear how many mice are tested per group, and how many independent experiments are performed.

Author's response: Additional details were added to the legends of Figure 3d-e, Figure 4b, Figure 5b, Figure 6c, Extended Data Figure 2, Extended Data Figure 3, Extended Data Figure 4, Extended Data Figure 6, so that the number of mice and the number of independent experiments are explicitly stated.

9. It appears that the sex of the mice was matched in each treatment group, however it is not clear overall how many male and female mice were tested in each treatment group.

Author's response: The numbers and sex of all experimental animals are indicated in the new document Extended Data Table 3. Each sheet represents a distinct experimental set-up and is named after the figure panel(s) on which the data appears.

('Minor comments':)

- Lines 311-326 of the discussion spend too much time re-summarising results.

Author's response: We have revised the referenced section, prioritizing brevity.

- Lines 330-332 should be expanded upon in the discussion.

Author's response: After rewriting the referenced Discussion section, our results are now better placed in scientific literature regarding the interplay between host diet and post-ampicillin microbiome recovery, see line 329-336.

- Lines 349-351 should also be expanded upon in the discussion.

Author's response: After rewriting the referenced Discussion section, we have now expanded on the relationship between direct bacterial competition and ampicillin degradation and their synergetic role in promoting *K. pneumoniae* decolonization, see line 351-361.

- Line 371-373: What was the minimum number of mice per cage?

Author's response: Each cage housed between 3-6 mice. We updated the Method section "Mice" to include this detail.

- Line 374: Should say sex not gender.

Author's response: The term "gender" was replaced with "sex" in the given line.

- Lines 380-384: The exact composition of the diets is not clear. What is in the standard chow diet? What else is in the high sucrose diet – presumably this is not 100% sucrose? Also, these diets described here do not match the descriptions of the diets tested in this paper (there was no fat only diet tested, and there is no description of the high starch diet)?

Author's response: The method section describing the dietary composition of the diets used was adjusted so that it does not suggest the use of a high-fat-only diet. Furthermore, the order number of the standard chow diet was added. Further details regarding the nutritional composition of all four diets were summarized in Extended Data Figure 4, which is displayed below.

Extended Data Table 4

	Standard chow	Synth. high-starch	Synth. high-sucrose	Synth. high-fat/high-sucrose
Gross Energy (GE)	16.2 MJ/kg	17.9 MJ/kg	17.9 MJ/kg	22.5 MJ/kg
Metabolizable Energy (ME)	13.5 MJ/kg	15.1 MJ/kg	15.2 MJ/kg	19.3 MJ/kg
Fat content (kJ%)	9	10	10	45
Protein content (kJ%)	24	20	20	20
Carbohydrates (kJ%)	67	70	70	35
Crude nutrients (%)				
Crude proteins (Nx6.25)	19	18.2	18.2	22
Crude fat	3.3	4.1	4.1	23.6
Crude fiber	4.9	5	5	5.7
Neutral Detergent Fiber	17.2			
Acid Detergent Fiber	7.1			
Crude ash	6.4	5.3	5.3	5.3
Starch	35.2	47	39.2	6.8
Sugar	5.3	1	16	21.1
N free extracts	54.2	62.9	63.2	40

- Lines 387-389: Are these patient isolates well characterized? Have the strains been deposited? Have their genomes been sequenced and has this sequencing data been deposited?

Author's response: The mentioned isolates were characterized in previous publications from our group (Osbelt et al., Cell Host Microbe 2021, Osbelt et al., Nature Microbiol 2024), the genomes were sequenced and made available under the following bioproject: <https://www.ebi.ac.uk/ena/browser/view/PRJEB42167>. For the *K. oxytoca* MK01 strain, a patent is pending (EP4259171A1, US20240041950A1); thus, it cannot be freely distributed. It can be provided upon agreeing to an MTA. Similarly, *K. pneumoniae* MD01 deposition is not permitted according to the corresponding ethical approval since patient consent does not cover depositing the strain. This strain can be requested from Prof. Dirk Schlüter. This information is included in the new section "Bacterial strains", line 385ff.

- Lines 398-400: Why were mice gavaged both orally and rectally, instead of just orally as is standard practice?

Author's response: When comparing the colonization efficiency of *Klebsiella oxytoca* MK01 following ampicillin treatment, we concluded that a combination of oral and rectal gavaging reaches more stable, consistent, and reliable colonization levels across all dietary backgrounds, thus avoiding the need for repeated gavaging in under-colonized groups.

- Line 435: Should this say M9 media not MM9 media? And was this performed aerobically with shaking?

Author's response: The text was modified to include additional technical details regarding aeration and shaking; furthermore, "MM9" was replaced with "M9".

- Shotgun metagenome sequencing data and LC-MS/MS data should be deposited into a public repository.

Author's response: Following the Reviewer's recommendation, we have uploaded our metagenome sequencing data to our previously created Bioproject: <https://www.ebi.ac.uk/ena/browser/view/PRJEB74255>.

Furthermore, we have deposited our LC-MS/MS data in the MassIVE repository (<https://massive.ucsd.edu/>). Prior to publication, the data can be reviewed using the following

credentials: username "MSV000095337_reviewer", password "a"; upon publication, the data will be made publicly accessible under the accession number MSV000095337.

- Line 535: Why was caffeine and trimethoprim measured?

Author's response: While no absolute quantification was done, caffeine and trimethoprim were used as standards in the extraction and resuspension solvent to monitor the reproducibility of the method and proper functioning of the equipment (e.g., to monitor that the desired volume of sample was consistently injected into the system).

Reviewer #3 (Remarks to the Author):

Reviewer #4 (Remarks to the Author):

*The authors aimed to recapitulate commensal-pathogen dynamics in a murine model of co-colonisation that is representative of westernised human diets of high starch, high sucrose, and high fat/high sucrose. The authors focus on the interaction between *Klebsiella oxytoca* and *Klebsiella pneumoniae* and employ a combination of amplicon 16S rRNA gene sequencing, metagenomic sequencing, mass spectrometry, and a suite of in vitro and in vivo experiments. I commend the authors on bringing together a well rounded study.*

I have both minor and major comments.

1. My main concern is with regards to the analysis of amplicon 16S rRNA gene data; namely, using QIIME1 for analysis here instead of QIIME2. QIIME1 is no longer supported and has been replaced by QIIME2 since the beginning of 2018. There are key changes between the two approaches in how microbial diversity is now calculated – i.e., calling amplicon sequence variants (ASVs) using QIIME2 which resolves single nucleotide variants, compared to clustering operational taxonomic units (OTUs) in QIIME1 at 97% similarity. This has implications in determining and quantifying compositional changes, and identifying key microbial taxa involved. Additionally, it was not clear if samples were rarefied prior to alpha and beta diversity calculations?

I would recommend recomputing microbiota analyses using QIIME2 and confirming similar conclusions can be made from this study before I would be comfortable recommending this manuscript for acceptance into Nature Communications. I think the quality of sequencing (i.e., using PE300) lends well to merging R1 and R2 reads for the V4 hypervariable region as the overlap here would be greater than 50bp, but it is not necessary to merge (c.f., <https://academic.oup.com/nar/article/35/18/e120/2401970>). When using QIIME2, I would suggest using deblur instead of dada2 (especially if keeping with merged reads) to determine ASVs.

Author's response: The reviewer's comment is correct, QIIME1 was replaced by QIIME2. At this point, our description was somewhat misleading: We only used the old version for some reformatting steps but already used the Usearch-pipeline for the important steps (error filtering, clustering). The Unoise3 algorithm performs similar to QIIME2-Deblur/dada2 to create OTUs/ASVs (see Prodan et al., Plos One 2020). Furthermore, all samples were rarefied as it is a standard feature of phyloseq. We still keep the read merging step as it is recommended for the Usearch-pipeline to address the typical lower quality of the R2 read. Nevertheless, after consideration of the Reviewer's comment regarding our amplicon 16S rRNA sequencing analysis, we decided to use an updated analysis pipeline avoiding the use of QIIME1 for any steps. The method section describing the technical details including specific commands and thresholds was re-written accordingly:

16S rRNA gene amplification and sequencing

Amplification of the 16S rDNA V4 hypervariable region (F515: GTGCCAGCMGCCGCGGTAA, R806; GGACTACHVGGGTWTCTAAT) was performed based on an established protocol previously described. In brief, input DNA for sequencing PCR was normalized to 25 ng/μl and performed with unique 12-base Golary barcodes incorporated via specific primers. After pooling and normalization to 10 nM, PCR amplicons were sequenced on an Illumina MiSeq platform via 250 bp paired-end sequencing (PE250). Resulting raw reads were demultiplexed by idmp (<https://github.com/yhmu/idemp>) according to the given barcodes. Libraries were processed, including merging the paired end reads, filtering the low-quality sequences, dereplication to find unique sequences, singleton removal, denoising, and chimera checking using the USEARCH pipeline 11.0.667. In brief, reads merged by fastq_mergepairs command (parameters: maxdiffs 30, pctid 70, minmergelen 200, maxmergelen 400), filtered for low quality with fastq_filter (maxee 1) and singletons using fastx_uniques command (minuniquesize 2). To predict biological sequences (ASVs, zOTUs) and filter chimeras we used the unoise3 command (minsize 10, unoise_alpha2), following the amplicon quantification using usearch_global command (strand plus, id 0.97, maxaccepts 10, top_hit_only, maxrejects 250). Taxonomic assignment was conducted by Constax (classifiers: rdp, syntax, blast) using the GreenGenes2 database and summarizing into biom-file for visualization in phyloseq and downstream analysis. Low abundance OTUs with less than 0.5% abundance in all samples were excluded. OTU table were resampled such that all samples have the same library size.

After reanalysis, the following figures were replotted, and statistical analysis was repeated (where applicable): Figure 4b-h, Extended Data Figure 2b-h, Extended Data Figure 3a-b, Extended Data Figure 4a-d.

Furthermore, the text referring to the exact relative abundances of certain taxa was also updated accordingly.

Some very minor corrections by editorial team throughout the text of the manuscript e.g.,

- Line 61: I think “both” is misused as only ‘systemic infection’ is provided.

Author’s response: The word “both” was removed from the sentence.

- Line 114: revise depended to dependent

Author’s response: The suggested correction was made.

- Line 196: sentence needs a rewrite: “However, HS diet showed a unique expansion of at Staphylococcaceae a group average of 37% in early post-ampicillin recovery.”

Author’s response: The sentence was reformulated the following way: “However, HS diet showed a unique expansion of Staphylococcaceae at a group average of 37% in early post-ampicillin recovery (Figure 4e-h).”

Line 309: revise to: “Amplicon 16S rRNA gene sequencing revealed...”

Author’s response: The term was revised in all of its occurrences throughout the manuscript.

- Line 409: revise to: “LB agar plates supplemented with 50 ug/ml kanamycin for *K. pneumoniae* selection and quantification, and MacConkey agar plates for *K. oxytoca* CFU quantification.”

Author's response: The suggested correction was made.

- Please clarify if referring to microbial composition or diet content in line 313: "Since the composition of the three semi-synthetic diets differs fundamentally in sucrose and fat content, the shared taxonomic changes are most probably attributed to the lack of complex dietary fibers in all three diets."
- As composition(al) is already being used to describe the microbial community, I would suggest revising line 313 to the effect of: "since the contents of the three semi-synthetic diets..."

Author's response: We clarified the wording, so that the statement explicitly refers to diet content rather than microbial composition:

"Since the nutrition content of the three semi-synthetic diets differs fundamentally in sucrose and fat content, the shared taxonomic changes are most probably attributed to the lack of complex dietary fibers in all three diets."

- Line 333: "Ex vivo assessment of intestinal content collected **from** ampicillin-treated mice with or without *K. oxytoca* pre-colonisation..."

Author's response: The suggested correction was made.

- Line 404: please provide the make and model of the mechanical lysis machine, and the value and units for the 'speed' setting.

Author's response: Further technical details for added to the text for clarification:

For in vivo quantification of *Klebsiella* strains from fecal samples:

"Samples were then homogenized with the BioSpec Mini-Beadbeater-96 once for 50 seconds at 2,400 rpm."

For Fecal microbial community DNA extraction:

"Samples were then homogenized with the BioSpec Mini-Beadbeater-96 three times for 5 minutes at 2,400 rpm; samples were kept on ice in between for 5 minutes to prevent overheating of the samples."

- Line 452: speed of mechanical lysis? And for all instances of mechanical lysis

Author's response: Please see answer to the previous comment for technical details.

- Line 454 and throughout: "Next, homogenized samples **were** centrifuged at 8,000 xg for 3 minutes."

Author's response: The correction was made in all of its occurrences throughout the manuscript.

- Line 431: what were the volume of cultures used?

Author's response: Further technical details were added to the text for clarification:

"After 16 hours bacterial lawns were scraped into PBS and adjusted to OD600 of 0.2, then inoculated in 1:10 ratio into M9 media with 5 g/L sucrose in a 96-well format under aerobic incubation with continuous shaking. OD600 was recorded every hour using a microplate reader by Biotek (Logphase600). Normalized values were visualized using software GraphPad Prism (version 8.2.1)."

- Line 454 and throughout: Consistency with using period or comma to denote numbers in the thousands.

Author's response: The text was revised to consistently use comma to denote numbers in the thousands.

- Line 469: please provide primer sequences.

Author's response: The sequences for primers F515 and R806 were added to the Methods under "16S rRNA gene amplification and sequencing"

- Line 469: please revise to read: "Amplification of the V4 hypervariable region of the 16S rRNA gene was performed..."

Author's response: The text was rephrased to "Amplification of the 16S rDNA V4 hypervariable region was performed..."

- Line 483: replace "OTU absolute abundance" with "OTU frequency table".

Author's response: The Method section "16S gene amplification sequencing" was re-written to reflect the updated data analysis pipeline. As a result, the text does not contain the term "OTU absolute abundance" anymore.

- It's great to see the amplicon 16S rRNA gene data being made publicly available. It would also be great to see the shotgun metagenomic and metabolomics data being made available too.

Author's response: Following the Reviewer's recommendation, we have uploaded our metagenome sequencing data to our previously created Bioproject: <https://www.ebi.ac.uk/ena/browser/view/PRJEB74255>

Furthermore, we have deposited our LC-MS/MS data in the MassIVE repository (MassIVE ID MSV000095337). Prior to publication, the data can be reviewed using the following credentials: username "MSV000095337_reviewer", password "a"; upon publication, the data will be made publicly accessible under the accession number MSV000095337.

- Extended data:
 - LefSe for MAGs and CAZymes have "???" in column headers – needs to be addressed

Author's response: The notes were removed from Extended data Table 1 and titles were added accordingly.

- Some references in text still read PMID

Author's response: All PMIDs were removed from the main text and the reference list was updated accordingly.

REVIEWER COMMENTS

Reviewer #1 (Remarks to the Author):

The authors have done an very good job addressing my feedback. The manuscript is now well-suited for publication in Nature Communications.

Response: Thank you for your support.

Reviewer #2 (Remarks to the Author):

I appreciate the steps the authors have taken to clarify points of their paper and to perform additional experiments to address some comments. However, unfortunately there are some fundamental aspects of this paper that I disagree with that I do not think can be addressed in a revised version of the manuscript:

*(1) The authors concluded that *K. oxytoca* protects the host by degrading ampicillin, allowing for recovery/protection of the gut microbiome, and it is gut microbiome recovery that protects against *K. pneumoniae* colonization. It is already well known that a healthy gut microbiome protects against *K. pneumoniae* intestinal colonization. Moreover, it is not a novel idea to degrade antibiotics to protect against pathogen intestinal colonization. Several previously published papers have explored direct administration of an enzyme to degrade several antibiotics (including more clinically relevant antibiotics than ampicillin) demonstrating that this can be accomplished in a safer manner than administering *K. oxytoca* (e.g. reviewed here: [*** The original URL has been rewritten. ***]). Finally, as mentioned in my previous comments, ampicillin is not relevant clinically to the group of patients that require new interventions to prevent or treat *K. pneumoniae* intestinal colonisation - immunocompromised adults. Therefore, I do not see the clinical relevance of *K. oxytoca* mediated protection described in this paper.*

*(2) The authors demonstrated that *K. oxytoca* is protective against *K. pneumoniae*, but as I mentioned in my previous comments an antibiotic-resistant opportunistic pathogen such as *K. oxytoca* will not be approved for clinical use in patient groups that are the most susceptible to the intestinal colonisation with *K. pneumoniae* – immunocompromised patients. I did not find the argument about *E. coli* to be convincing, as *E. coli* is often found in healthy adults unlike *K. oxytoca*. Moreover, the argument that *K. oxytoca* can asymptotically colonize infants and children is not reassuring, as another pathogen *Clostridioides difficile* can also asymptotically colonize infants but can cause disease in adults. So again, I don't see the clinical relevance of using *K. oxytoca* to prevent *K. pneumoniae* intestinal colonization, especially if safer alternatives such as engineering a commensal to degrade the antibiotic or using purified enzymes directly (which is already being explored as mentioned above) are available.*

Response: As instructed by the editor, we have added a statement explaining that *K. oxytoca* is an opportunistic pathogen, and that further work would be needed to fully evaluate its safety (in line with their second concern). This statement can be found in Lines 348 - 355

Reviewer #3 (Remarks to the Author):

I co-reviewed this manuscript with one of the reviewers who provided the listed reports. This is part of the Nature Communications initiative to facilitate peer review training and provide appropriate recognition for Early Career Researchers who co-review manuscripts.

Reviewer #4 (Remarks to the Author):

major:

I appreciate the authors efforts in addressing the QIIME1 vs., QIIME2 workflow. I believe the initial concerns have been addressed.

minor:

reviewer comment version 1, Line 469: please revise to read: “Amplification of the V4 hypervariable region of the 16S rRNA gene was performed...”

Author’s response: The text was rephrased to “Amplification of the 16S rDNA V4 hypervariable region was performed...”

16S rDNA V4 is incorrect terminology. Should read “16S rRNA gene”, which is at the DNA level.

i.e., “Amplification of the 16S rRNA gene V4 hypervariable region was performed...”

Response: We have corrected the error. Thank you for noticing.